# Multilineage murine stem cells generate complex organoids to model distal lung development and disease

Ana Ivonne Vazquez-Armendariz[1] [ID], Monika Heiner[1], Elie El Agha[1] [ID], Isabelle Salwig[2], Andreas Hoek[3], Marie Christin Hessler[1], Irina Shalashova[1], Amit Shrestha[1], Gianni Carraro[4], Jan Philip Mengel[5], Andreas Günther[1], Rory Edward Morty[1,6] [ID], István Vadász[1], Martin Schwemmle[7,8] [ID], Wolfgang Kummer[9], Torsten Hain[5], Alexander Goesmann[3], Saverio Bellusci[1], Werner Seeger[1,6], Thomas Braun[2] [ID] & Susanne Herold[1,*] [ID]

## Abstract

Organoids derived from mouse and human stem cells have recently emerged as a powerful tool to study organ development and disease. We here established a three-dimensional (3D) murine bronchioalveolar lung organoid (BALO) model that allows clonal expansion and self-organization of FACS-sorted bronchioalveolar stem cells (BASCs) upon co-culture with lung-resident mesenchymal cells. BALOs yield a highly branched 3D structure within 21 days of culture, mimicking the cellular composition of the bronchioalveolar compartment as defined by single-cell RNA sequencing and fluorescence as well as electron microscopic phenotyping. Additionally, BALOs support engraftment and maintenance of the cellular phenotype of injected tissue-resident macrophages. We also demonstrate that BALOs recapitulate lung developmental defects after knockdown of a critical regulatory gene, and permit modeling of viral infection. We conclude that the BALO model enables reconstruction of the epithelial–mesenchymal-myeloid unit of the distal lung, thereby opening numerous new avenues to study lung development, infection, and regenerative processes *in vitro*.

**Keywords**  BALO; BASC; lung organoids; stem cells
**Subject Categories**  Stem Cells & Regenerative Medicine; Respiratory System
**The EMBO Journal (2020) 39: e103476**

## Introduction

In recent years, three-dimensional (3D) organoid systems derived from mouse and human adult stem cells or induced pluripotent stem cells (iPSC) have emerged as a powerful technology for *in vitro* modeling of organ development and disease (Lancaster & Knoblich, 2014; Kretzschmar & Clevers, 2016). Adult somatic tissue-resident stem/progenitor cells represent an excellent starting point to generate 3D organoid systems due to their ability to proliferate and differentiate into cell types found in the corresponding parental tissues. Murine lung organoids mimicking—to different extents—the cellular morphology and certain functional features of the native organ have been generated from tissue-resident cells by growth factor supplementation or by co-culture with feeder cells (Rock *et al*, 2009; Chen *et al*, 2012a; Barkauskas *et al*, 2013; Lee *et al*, 2014). For instance, single basal cells give rise to organoids termed "tracheospheres" consisting of basal and ciliated luminal cells (Rock *et al*, 2009). Type 2 alveolar epithelial cells (AEC II) co-cultured with PDGFRα$^+$ lung mesenchymal cells form alveolar-like structures consisting of type 1 alveolar epithelial cells (AEC I) and AEC II (Chen *et al*, 2012a; Barkauskas *et al*, 2013). Furthermore, co-culture of multipotent lung epithelial stem cells with endothelial cells gives rise to various organoid structures (Lee *et al*, 2014).

However, all 3D organoids described so far only recapitulate limited morphological and cellular features of the lung. A model that recapitulates the 3D complexity of the bronchioalveolar

1  Department of Internal Medicine II and Cardio-Pulmonary Institute (CPI), Universities of Giessen and Marburg Lung Center (UGMLC), Member of the German Center for Lung Research (DZL) and The Institute of Lung Health (ILH), Giessen, Germany
2  Department of Cardiac Development and Remodelling, Max Planck Institute for Heart and Lung Research, Bad Nauheim, Germany
3  Bioinformatics and Systems Biology, Justus-Liebig-University, Giessen, Germany
4  Lung and Regenerative Medicine Institutes, Cedars-Sinai Medical Center, Los Angeles, CA, USA
5  Institute for Medical Microbiology, German Center for Infection Research (DZIF), Justus-Liebig-University Giessen, Partner Site Giessen-Marburg-Langen, Giessen, Germany
6  Department of Lung Development and Remodelling, Max Planck Institute for Heart and Lung Research, Bad Nauheim, Germany
7  Institute of Virology, Medical Center—University of Freiburg, Freiburg, Germany
8  Faculty of Medicine, University of Freiburg, Freiburg, Germany
9  Institute for Anatomy and Cell Biology, Justus-Liebig-University, UGMLC, DZL, Giessen, Germany
   *Corresponding author. E-mail: Susanne.Herold@innere.med.uni-giessen.de

compartment of the lung is critically missing. Current lung organoid models mostly consist of only epithelial cells, and in rare cases, the mesenchymal compartment. Nonetheless, tissue-resident cells of myeloid origin, which are important mediators of lung development, immune response, and tissue regeneration, remain absent. We have previously described an epithelial cell population of the distal lung that is characterized by the signature EpCAM$^{high}$CD24$^{low}$Sca-1$^+$ and is enriched with epithelial stem/progenitor cells that exhibit high proliferative potential in response to influenza A virus (IAV)-induced lung injury. We also showed that cells within this population proliferate and form organospheres in Matrigel cultures in the presence of fibroblast growth factor 10 and hepatocyte growth factor. Following orthotopic transplantation into IAV-injured murine lungs, EpCAM$^{high}$CD24$^{low}$Sca-1$^+$ cells regenerate the distal lung tissue and give rise to AEC I, thus demonstrating inherent stem cell properties (Quantius et al, 2016). Accordingly, bronchioalveolar stem cells (BASCs) represent an important stem/progenitor cell population that co-expresses SCGB1A1 and SFTPC and can differentiate into club cells, AECII, and AECI in vitro (Giangreco et al, 2002; Lee et al, 2014). Our recently published data showed that BASCs contribute to regeneration of both bronchiolar epithelium and alveolar epithelium following lung injury including influenza virus-induced damage in vivo (Salwig et al, 2019).

In the present study, we isolated an epithelial cell population that contains BASCs to develop a robust protocol for generation of bronchioalveolar lung organoids (BALOs) comprised of spatially organized bronchiolar-like and alveolar-like structures after co-culture with defined subsets of lung-resident mesenchymal cells (rMC). Within the organoids, cells in the alveolar-like regions differentiated into mature AEC II and AEC I, whereas the airway-like regions contained basal cells, secretory and ciliated cells. In addition, we provide evidence for successful engraftment of tissue-resident, yolk sac-derived alveolar macrophages (TR-Mac) into the developing BALO. We also demonstrate the usefulness of our 3D model to study lung development by inhibition of microRNA 142-3p (miR-142-3p) gene expression. Finally, we show that BALOs can be infected with IAV, which opens new avenues to study and visualize processes of lung infection, injury, and repair.

## Results

### 3D co-culture of BASC and rMC results in the formation of BALO

To establish lung organoids, rMC (defined as EpCAM$^-$Sca-1$^+$) and epithelial stem/progenitor cells were isolated from leukocyte/endothelial cell-depleted lung homogenates of wild-type (WT) mice by FACS. The latter population was selected according to a previously defined signature enriching epithelial stem/progenitor cells, using the surface markers EpCAM$^{high}$CD24$^{low}$Sca-1$^+$ (Quantius et al, 2016; Fig 1A). Flow-sorted EpCAM$^{high}$CD24$^{low}$Sca-1$^+$ cells in combination with rMC were seeded into Matrigel, and the development of organoid structures was monitored for at least 21 days. EpCAM$^{high}$CD24$^{low}$Sca-1$^+$ cells expanded and formed colonies which developed into organospheres as illustrated in Fig 1B. After 10–11 days in culture, lung organoids started to develop buds that

formed well-defined central bronchiolar-like structures and peripheral alveolar-like structures by day 21 (Fig 1B and Movie EV1). Confocal microscopy showed that bronchiolar-like structures within BALO were positive for the club cell marker secretoglobin family 1 A member 1 (SCGB1A1), whereas surrounding alveolar-like regions stained positive for the AEC II marker surfactant protein C (SFTPC) (Appendix Fig S1A). In addition to BALOs, two other types of organoids formed, albeit at a far lower frequency. These structures resembled the previously described bronchiolar organoids composed of large tubes (bronchiolospheres) and alveolospheres characterized by a compact saccular arrangement (Appendix Fig S1B; Barkauskas et al, 2013; Lee et al, 2014). Our findings suggested that the EpCAM$^{high}$CD24$^{low}$Sca-1$^+$ population used for organoid formation was not homogeneous but contained multilineage-committed progenitor cells with bronchial epithelial cell and AEC differentiation potential. Quantitative analysis showed that about 80% of organoids formed per well were BALOs in an initial culture (passage 0; P0), whereas 6 and 14% revealed an alveolar and bronchiolar phenotype, respectively (Appendix Fig S1C). To verify that BALOs were generated by clonal expansion of a putative stem cell rather than by self-assembly of several cell types contained within the EpCAM$^{high}$CD24$^{low}$Sca-1$^+$ population, cells expressing membrane-targeted tdTomato were cultured with a mixture of GFP-expressing cells (Fig 1C). As would be expected for organoids derived from a single cell, only GFP$^+$ or tdTomato$^+$ organoids were observed but no mixed phenotypes. Taken together, these data highlight the potential of single EpCAM$^{high}$CD24$^{low}$Sca-1$^+$ cells to proliferate and differentiate into a complete lung organoid in the presence of rMC.

To further define the stem/progenitor cell phenotype within the EpCAM$^{high}$CD24$^{low}$Sca-1$^+$ cell population, additional known epithelial stem cell markers were analyzed. BASCs are located at the bronchoalveolar duct junction and have been shown to be able to expand after injury, giving rise to differentiated alveolar as well as bronchiolar cells (Kim et al, 2005; Liu et al, 2019; Salwig et al, 2019). BASCs co-express the club cell marker SCGB1A1 and the AEC II marker SFTPC. We used EpCAM$^{high}$CD24$^{low}$Sca-1$^+$ cells from Scgb1a1$^{mCherry}$Sftpc$^{YFP}$ double reporter mice (SPC$^{-2A-YFP-2A-tTA-N}$, CCSP$^{-2A-mCherry-2A-tTA-C}$) to determine mCherry and YFP co-expression within the EpCAM$^{high}$CD24$^{low}$Sca-1$^+$ population (Salwig et al, 2019). About 5% of the cells were double-positive for SCGB1A1 and SFTPC (Appendix Fig S1D). While most cells (95%) were SCGB1A1$^+$SFTPC$^-$, SCGB1A1$^-$SFTPC$^+$ cells were detected at a very low level in this fraction (< 0.5%). Although SCGB1A1$^+$SFTPC$^+$ cells were present at low frequency, mature (mCherry$^+$YFP$^+$) BALOs represented 80% of all organoids at day 21 (Fig 1D). Bronchiolospheres (composed solely of SCGB1A1$^+$ cells) accounted for only 11%, whereas alveolospheres (composed solely of SFTPC$^+$ cells) accounted for 9% of all organoids. To address the respective cell of origin of the different organoid phenotypes, EpCAM$^{high}$CD24$^{low}$Sca-1$^+$ cells were flow-sorted according to single or double expression of SCGB1A1 and SFTPC, and developing organoids were followed over time by confocal microscopy. Culture of SFTPC$^+$ or SCGB1A1$^+$ single-positive EpCAM$^{high}$CD24$^{low}$Sca-1$^+$ cells resulted in the formation of alveolospheres or bronchiolospheres, respectively, while the formation of BALOs was not detected in cultures with either of these two populations (Appendix Fig S1E and F). SCGB1A1$^+$SFTPC$^+$ double-positive cells gave rise to BALOs at a

colony-forming frequency of 1:100, indicating their BASC phenotype (Fig 1E and Appendix Fig S1G). Given that 95% of EpCAM$^{lowSFTPC+}$ AEC II (Quantius *et al*, 2016) are excluded by the EpCAM$^{highCD24low}$Sca-1$^+$ gating strategy, AEC II were isolated based solely on SFTPC single expression to evaluate their potential to generate organoids under our culture conditions. The resulting organoids formed at a colony-forming frequency of 1:25 from SFTPC$^+$ cells and were exclusively of alveolospheres phenotype, whereas isolation of only SCGB1A1$^+$ cells without pre-enrichment showed a colony-forming frequency of 1:1,000 and generated bronchiolospheres (Appendix Fig S1G). We then used EpCAM$^{high}$CD24$^{low}$Sca-1$^+$ cells from *Scgb1a1$^{mCherry}$Sftpc$^{YFP}$* double reporter mice to determine mCherry and YFP expression during BALO formation. Within the early-stage BALO, SCGB1A1$^+$ and SFTPC$^+$ cells were uniformly distributed with only a small fraction of double-positive cells remaining at day 8 of culture. Day 21 BALOs were comprised of SCGB1A1$^+$ central branches surrounded by SFTPC$^+$ alveolar-like structures (Figs 1E and EV1), confirming the cellular and structural composition of BALO at this later stage (Appendix Fig S1A). In addition, we used "BASC v-race" (SPC$^{-2A-YFP-2A-tTA-N}$, CCSP$^{-2A-mCherry-2A-tTA-C}$, tetO$^{biluc/Cre}$, Rosa26$^{stopflox-lacZ}$) mice in which BASCs and all their descendants are permanently labeled via β-galactosidase activity to generate organoids. After 21 days of culture, completely LacZ$^+$ BALOs were observed indicating that all cells in BALO originate from BASC (Appendix Fig S1H). Finally, to address whether BALOs still contained stem/progenitor cell pools, we digested only BALOs from the initial (P0) culture by manually removing all bronchiolospheres and alveolospheres and re-cultured them in the presence of freshly sorted rMC (P1). As shown in Appendix Fig S1C, new BALOs formed in P1 cultures, although with a lower frequency as in the initial culture (19%), suggesting that within BALO there are progenitor cells capable of forming bronchiolospheres and alveolospheres, such as SCGB1A1$^+$ bronchiolar and SFTPC$^+$ alveolar progenitor cells. Nonetheless, BALOs were still formed after further passaging of these mixed organoids indicating the presence of BASCs within BALO (Appendix Fig S1C). Moreover, WT rMC were co-cultured with BASCs that were isolated from the lungs of "BASC viewer" (SPC$^{-2A-YFP-2A-tTA-N}$, CCSP$^{-2A-mCherry-2A-tTA-C}$, tetO$_{bi}$$^{lacZ/huGFP}$) mice, which harbor a split-tTA construct at the endogenous gene loci to allow the identification of SCGB1A1$^+$SFTPC$^+$ double-positive cells via β-galactosidase labeling (Salwig *et al*, 2019). Notably, LacZ staining of the cultures revealed the presence of SCGB1A1$^+$SFTPC$^+$ BASCs within BALO, located at the distal regions even after 60 days of culture (Appendix Fig S1I).

To analyze the cellular composition of BALO at a mature stage, single-cell RNA sequencing (scRNA-Seq) was performed on digested day 21 BALOs. Data analysis revealed the presence of four distinct clusters including two epithelial (C1 and C2) and two mesenchymal subpopulations (C3, myofibroblasts (MYO); and C4, matrix fibroblasts/lipofibroblasts (LIFs)) (Fig 1F). Cells in the epithelial clusters expressed both airway- and alveoli-associated genes (Fig 1G and H). Airway-associated genes (C1) identified cellular markers for ciliated cells (*Itgb4*), basal cells (*Trp63, Krt7*), and respiratory epithelium (*Sox2*) (Fig 1G; Treutlein *et al*, 2014; Du *et al*, 2015, 2017). The alveolar cluster (C2) included genes described as lineage markers for AEC II such as *Cxcl15, Lyz2,* and *Sftpc,* and AEC I lineage markers such as *Hopx* (Fig 1H; Treutlein *et al*, 2014; Du *et al*, 2015, 2017).

## The BALO model mimics the 3D morphology and cellular composition of the bronchioalveolar compartment with proximo-distal patterning and pulmonary surfactant secretion

To further characterize and validate the cellular composition of BALO, day 0 and day 21 cultures were digested with dispase and analyzed by FACS (Appendix Fig S2A). Corresponding to the scRNA-Seq data, BALOs were composed of an EpCAM$^+$ epithelial and EpCAM$^-$ non-epithelial fraction. The EpCAM$^+$ BALO population consisted of cell populations that express typical markers of differentiated airway and alveolar epithelium. These subpopulations included a major fraction of AEC II (94.3 ± 0.2%), low-frequent podoplanin (PDPN)$^+$ AEC I (5.3 ± 0.6%), and EpCAM$^{high}$CD24$^{high}$ small airway (bronchial) epithelial cells (Appendix Fig S2A). Furthermore, we validated upregulation of lineage markers of differentiated cell types of the adult lung by qPCR. In day 21 BALO cultures, upregulated genes included *Hopx* for AEC I and *Sftpc* for AEC II, as well as *Foxj1, Muc5ac,* and *p63* for ciliated, goblet cells, and basal cells, respectively (Fig 2A).

Moreover, electron microscopy analyses revealed that alveoli were lined with a single layer of epithelial cells with short microvillous protrusions. These cells were interconnected by tight junctions and contained abundant mitochondria and numerous lamellar bodies, which represent a characteristic feature of mature AEC II (Fig 2B). Notably, bronchoalveolar duct junction-like regions containing bronchiolar and intermediate cells types that lead into alveolar-like regions with flattened AEC I were identified within BALO (Appendix Fig S2B). Cells lining the airway-like tubes were devoid of surfactant and showed different phenotypes from undifferentiated epithelial cells to ciliated airway cells found to be interconnected by typical junctional complexes at their apical surfaces (Fig 2C). To demonstrate the presence of surfactant production, BALOs were stained with LysoTracker, which has been shown to accumulate in lamellar bodies, and LipidTOX for phospholipids as a major component of pulmonary surfactant. As expected, alveolar-like areas within BALO showed the presence of lamellar bodies and phospholipids (Fig 2D and Appendix Fig S2C). Furthermore, surfactant production in BALO was confirmed by Western blot analysis of pro-SPC, mature SPC, pro-SPB, mature SPB, and SPA (Fig 2E). Further proximo-distal specification of BALO was demonstrated by identification of PDPN$^+$ AEC I within the alveolar-like regions in BALO generated from *Pdpn*$^{GFP}$ reporter mice and by staining of β-IV tubulin$^+$ ciliated cells located in the airway-like sections (Fig EV2A and B).

A recent stereological analysis has shown that the mean diameter of murine lung alveoli decreases during the first weeks after birth, while the number of alveoli increases before reaching a plateau throughout adulthood (Pozarska *et al*, 2017). In this regard, analysis of BALO distal structures at different stages revealed that the dynamic changes in terms of size and number followed a similar trend during BALO formation. The mean diameter of the distal alveolar-like structures decreases by 23% between days 15 and 21, 13% between days 21 and 30, and remains unchanged until day 40. Correspondingly, BALOs showed a threefold increase in the mean number of alveolar-like structures from day 15 to 30 (Fig 2F and G, and Appendix Fig S2D).

Notably, alveolar-like structures within BALO showed a lumen filled with lamellar surfactant and lined by thin and elongated AEC I

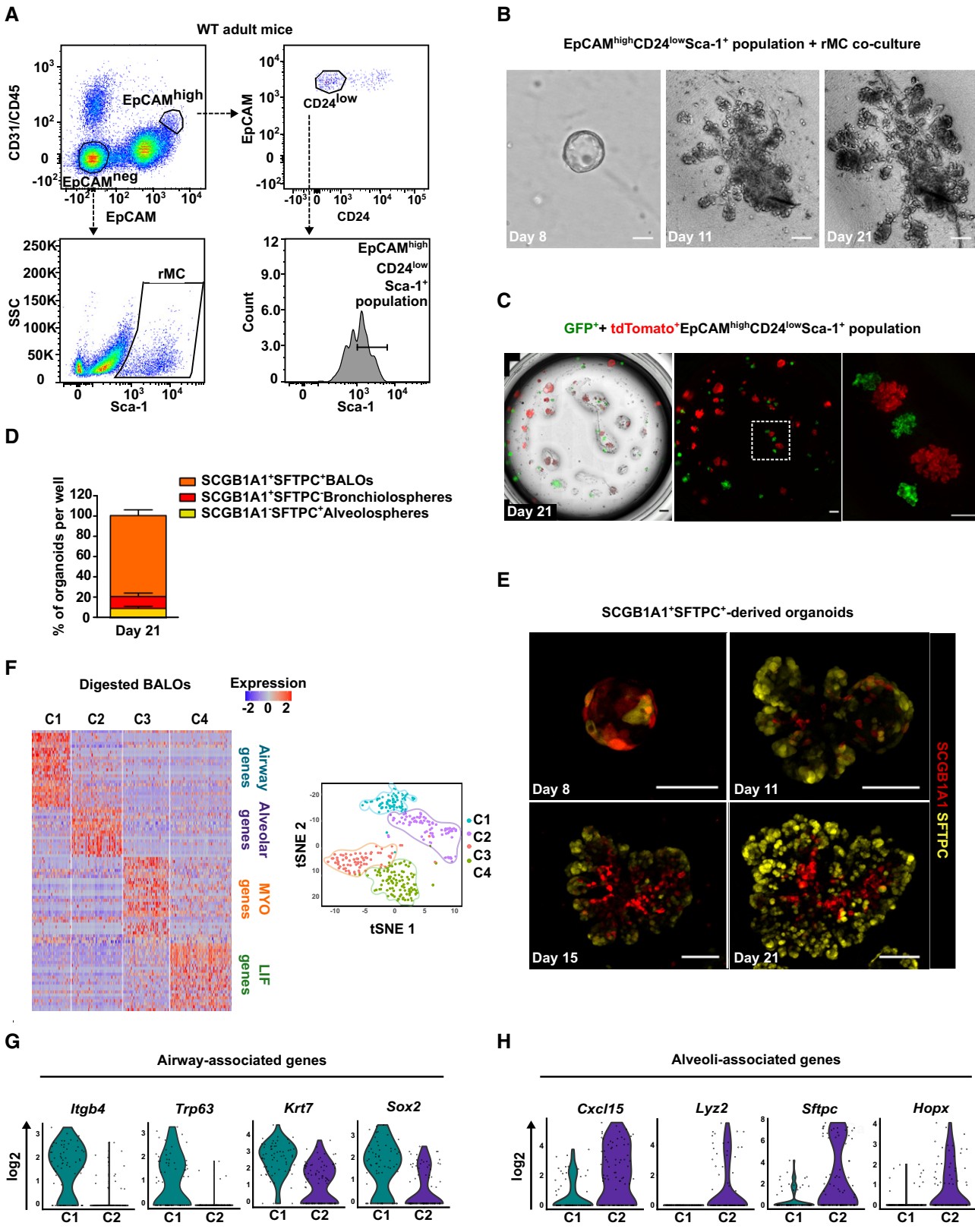

**Figure 1.**

**Figure 1.  3D co-culture of BASC and rMC results in the formation of BALO.**

A       Gating strategy for sorting of EpCAM$^{high}$CD24$^{low}$Sca-1$^{+}$ cells and rMC from lung homogenate of adult mice.

B       Time course of epithelial stem/progenitor cell proliferation and differentiation within BALO at days 8, 11, and 21 of co-culture with rMC.

C       Clonal expansion of EpCAM$^{high}$CD24$^{low}$Sca-1$^{+}$ cells derived from either GFP-expressing mice or tdTomato-expressing mice at day 21 of co-culture.

D       Percentages of SCGB1A1$^{+}$SFTPC$^{+}$ (BALOs), SCGB1A1$^{+}$SFTPC$^{-}$ (bronchiolospheres), and SCGB1A1$^{-}$SFTPC$^{+}$ (alveolospheres) organoids per well at day 21 of culture derived from EpCAM$^{high}$CD24$^{low}$Sca-1$^{+}$ cells isolated from Scgb1a1$^{mCherry}$Sftpc$^{YFP}$ reporter mice (n = 4 biological replicates).

E       Representative confocal images of days 8–21 of culture showing endogenous SCGB1A1 and SFTPC expression during BALO formation derived from EpCAM$^{high}$CD24$^{low}$Sca-1$^{+}$SCGB1A1$^{+}$SFTPC$^{+}$ cells isolated from Scgb1a1$^{mCherry}$Sftpc$^{YFP}$ reporter mice.

F–H     Heat map (left) and tSNE plot (right) (F) of digested day 21 BALO cultures depicting four distinct clusters (C1, airway, blue; C2, alveolar, purple; C3, MYO, orange; and C4, LIF, green). Violin plots of selected genes representing airway-associated genes (G) (*Itgb4, Trp63, Krt7,* and *Sox2*) and alveoli-associated genes (H) (*Cxcl15, Lyz2, Sftpc,* and *Hopx*). Each violin plot shows the frequency distribution of the mean transcript level (log$_2$).

Data information: Scale bars = 100 μm (B, C right, E) or 500 μm (C left). Bar charts presented as the mean ± SEM.

intercalated by cuboidal AEC II joined through tight junctions after 40 days of culture (Fig 2H and Appendix Fig S2E). The BALO airway-like ultrastructure forms a pseudostratified epithelium comprised of basal-like cells, differentiated secretory cells filled with secretory granules, and ciliated cells with mature cilia and basal bodies aligned underneath the apical cell surface. Of note, kinocilia with the classical central microtubule pair (9 × 2 + 2 configuration) could be identified in micrographs with higher magnification (Fig 2I).

Taken together, our *in vitro* model meets criteria that identify organoids based on the definition published by Lancaster and Knoblich, including (i) composition of multiple organ-specific cell types, (ii) recapitulation of specific organ features such as lumen-directed secretion of pulmonary surfactant by AEC II, and, importantly, (iii) spatially restricted cell lineage commitment with organization into defined airway- and alveolar-like compartments with clear proximo-distal patterning (Lancaster & Knoblich, 2014).

### Distinct subsets of EpCAM$^{-}$Sca-1$^{+}$ rMC-derived fibroblasts are indispensable for BALO growth, cell differentiation, and branching morphogenesis, and model the mesenchymal niche of the lung

Having characterized the BALO epithelial compartment, we sought to phenotype the mesenchymal compartment during BALO development and after differentiation. EpCAM$^{-}$Sca-1$^{+}$ of non-leukocyte and non-endothelial origin (Fig 1A, rMC) was previously found to support lung organoid formation (Quantius *et al*, 2016). In accordance with our scRNA-Seq analysis and previously published data showing that rMC are a heterogeneous population including progenitors of MYO and LIF (Fig 1G; Perl & Gale, 2009; Al Alam *et al*, 2015), we found by microscopic analysis of BALO cultures that rMC gave rise to at least two distinct fibroblast cell types (Fig 3A and B, and Appendix Fig S3A). One population contained LipidTOX-positive lipid bodies, characteristic of LIF, while the other population consisted of alpha-smooth muscle actin (αSMA)-positive, spindle-shaped cells with long cellular extensions and ample cisterns of rough endoplasmic reticulum resembling MYO (Figs 2B and 3B, and Appendix Fig S3B).

It has been reported that MYO and LIF are defined by the differential expression of platelet-derived growth factor receptor alpha (PDGFRα) and αSMA (Perl & Gale, 2009). Using a *Pdgfra$^{GFP}$* knock-in mouse line, two populations of rMC necessary for BALO formation were defined: a Sca-1$^{high}$PDGFRα$^{low}$ and a Sca-1$^{int}$PDGFRα$^{high}$ fraction (Fig 3C and D). Intriguingly, although colony-formation efficiency was the same as when both PDGFRα populations were present, Sca-1$^{high}$PDGFRα$^{low}$ rMC allowed early organoid growth but were unable to support alveolar differentiation or sustain branching morphogenesis (Appendix Fig S3D and E). In contrast, co-culture of BASCs with Sca-1$^{int}$PDGFRα$^{high}$ rMC did not support organoid outgrowth, suggesting that both populations were required for complete BALO formation (Fig 3D). To verify that Sca-1$^{int}$PDGFRα$^{high}$ rMC were necessary to drive branching morphogenesis, BALO cultures grown in the presence of Sca-1$^{high}$PDGFRα$^{low}$ rMC fraction were complemented with Sca-1$^{int}$PDGFRα$^{high}$ rMC at day 7 of culture (shortly before first branches start to form in BALO). Branching morphogenesis could be rescued to a large extent; however, BALO formation was found to be delayed and did not reach BALO full size until day 28. These data indicate that Sca-1$^{int}$PDGFRα$^{high}$ rMC are essentially driving branching and that an early crosstalk between both rMC populations and BASCs might be essential for proper dynamics of BALO formation (Appendix Fig S3C).

Using our scRNA-Seq data set (Fig 1G), we revealed two mesenchymal cell clusters. Among the genes expressed within these clusters are genes previously associated with MYO (C3) and LIF (C4) phenotypes including *Fgf10, Pdgfrα, Tagln, Acta2,* and *Eln* genes (Fig 3E and Table 1; Perl & Gale, 2009; McGowan & McCoy, 2014; Al Alam *et al*, 2015), thus confirming our morphological data. Of note, PDGFRα expression was higher in MYO than in LIF, in line with previous findings on these mesenchymal cell subsets in our BALO model (Perl & Gale, 2009). LIF-associated genes previously defined by RNA-Seq analysis of lung samples such as *Apoe, Serpina3a, Gsn,* and *Gas6* were detected in the LIF cluster (Du *et al*, 2015, 2017). In addition, *Axin2,* a gene recently related to the MYO phenotype in the lung, was also expressed in the BALO MYO cluster (Zepp *et al*, 2017).

In the developing lung, lipid-droplet-containing LIFs are found in close proximity to the alveolar epithelium promoting epithelial growth and AEC II differentiation (El Agha *et al*, 2017). Moreover, LIFs allow primary murine AEC to form alveolospheres under co-culture conditions by providing growth factors (Barkauskas *et al*, 2013). In accordance, we detected LipidTOX$^{+}$ LIF frequently distributed around developing organospheres and around BALO. LipidTOX$^{+}$ LIFs were found to be PDGFRα$^{low}$, whereas αSMA$^{+}$ alveolar MYOs were confined to the PDGFRα$^{high}$ expressing fraction (Fig 3F). MYO had been demonstrated to deposit extracellular matrix in the neonatal lung, thereby providing the necessary scaffold for alveolarization (El Agha & Bellusci, 2014). Consistently, PDGFRα$^{high}$ MYO accumulated at branching sites within the center of the BALO (Fig 3G).

In summary, our data revealed the presence of distinct subsets of PDGFRα high- and low-expressing MYO and LIF required for BALO formation. MYO and LIF are spatially organized within or in close proximity to alveolar-like structures. Therefore, BALO might be useful for modeling the pulmonary mesenchymal niche and for studying of epithelial–mesenchymal crosstalk under different conditions.

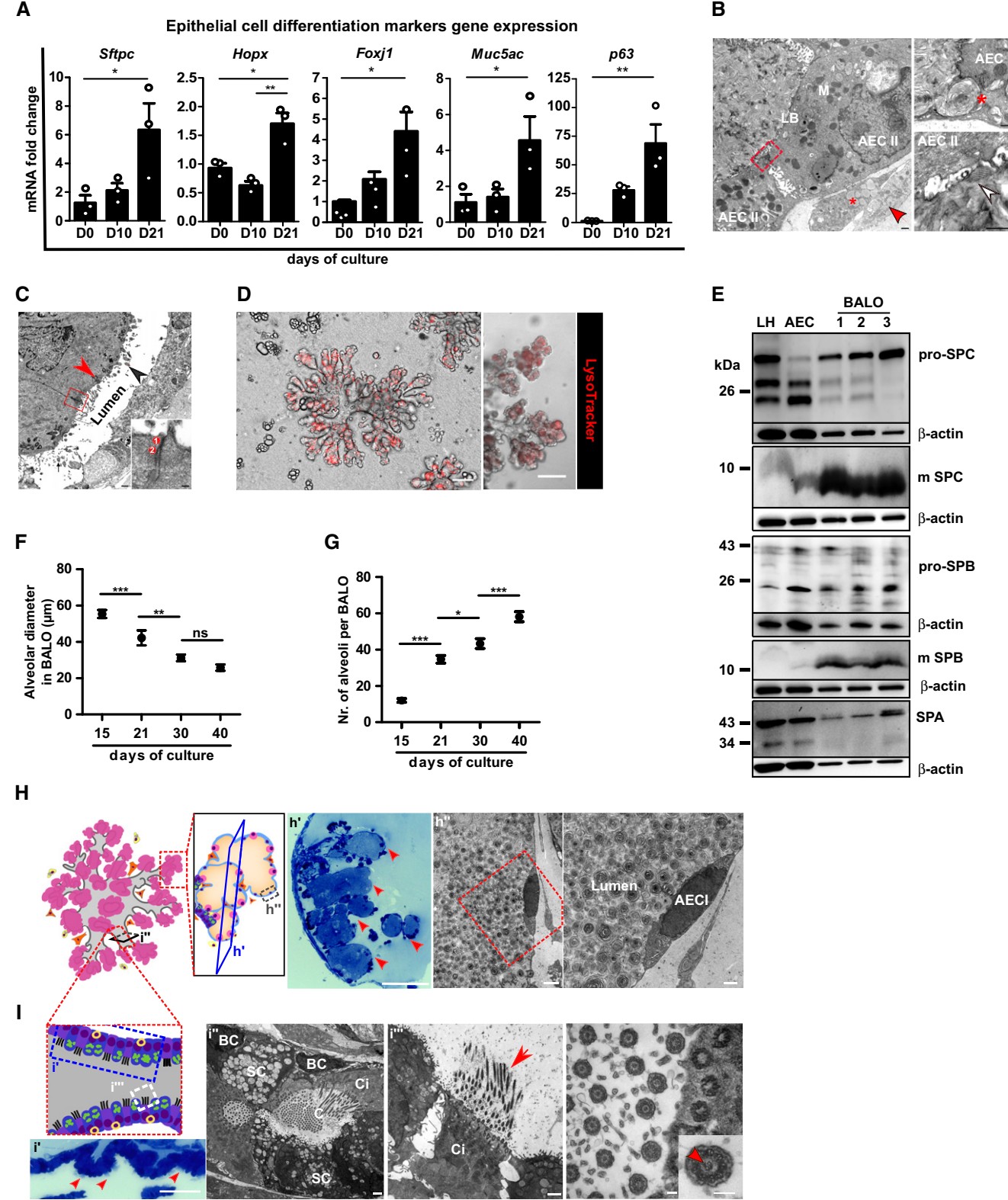

Figure 2.

**Figure 2. The BALO model mimics the 3D morphology and cellular composition of the bronchioalveolar compartment with proximo-distal patterning and pulmonary surfactant secretion.**

A  mRNA expression analysis of epithelial cell differentiation markers *Sftpc* (AEC II), *Hopx* (AEC I), *Foxj1* (ciliated cells), *Muc5ac* (secretory cells), and *p63* (basal cells) in BALO at days 0, 10, and 21 of culture (n = 3 biological replicates with pooled cells from 4 cultures per replicate).

B  Electron microscopy of BALO alveoli showing two cuboidal epithelial cells (AEC II) connected by tight junctions (red square) with numerous mitochondria (M) and lamellar bodies (LB). A LIF with numerous lipid droplets (*) and a MYO with cisterns of rough ER (red arrow) are located at the basal side (left). The alveolar lumen is filled with lamellar surfactant including tubular myelin (white arrow) (lower right). Exocytosis of a lamellar body (*) from an AEC II (upper right). Scale bars indicate 500 nm.

C  Electron microscopy of bronchiolar-like airway depicting columnar ciliated cells (red arrowhead) with basal bodies (black arrowhead) at the left side of the longitudinally sectioned lumen. The boxed area indicates an apical junctional complex between two ciliated cells: 1 = tight junction and 2 = adherens junction. Scale bar indicates 500 nm (in insert: 100 nm).

D  Staining of lamellar bodies in BALO with RFP LysoTracker. Scale bars represent 100 μm.

E  Western blot analysis of the surfactant proteins: pro-SPC, mature SPC, pro-SPB, mature SPB, and SPA in lung homogenate (LH), AEC, and day 21 BALO (n = 3 biological replicates).

F, G  Alveolar diameter (F) and number of alveoli (G) in tdTomato+ BALO at days 15, 21, 30, and 40 of culture were measured from n = 5 BALOs in n = 3 biological replicates BALOs.

H, I  Representative scheme and images of day 40 BALO alveoli (H) and airway (I). BALO alveolar-like structures (H) are shown (h′) (red arrowheads) in semi-thin section (0.5 μm) stained with Toluidine blue. Scale bar indicates 100 μm (left). Electron microscopy showing AEC I (h″) ultrastructure within BALO. Scale bars indicate 2,500 nm (center) and 1,000 nm (right). An airway-like structure (i′) is shown with secretory and ciliated cells (red arrowheads) in semi-thin sections (0.5 μm) longitudinally cut and stained with Toluidine blue. Scale bar indicates 50 μm (far left). Electron microscopy of a bronchiolar-like airway (I) depicting pseudostratified epithelium (i″) with a basal-like cell (BC), not reaching the lumen in which cilia (C) are seen, located between a secretory (SC) and a ciliated cell (Ci). Scale bars indicate 1,000 nm (left and right). Mature cilia (i‴) in BALO at higher magnification depicting the 9 × 2 + 2 structure with central doubled microtubules (insert, red arrowhead), a characteristic for motile cilia. Scale bar indicates 100 nm (far right).

Data information: Bar and dot charts presented as the mean ± SEM and probability determined using one-way ANOVA (*$P < 0.05$, **$P < 0.01$, ***$P < 0.001$).

## TR-Mac engraft into BALO alveolar-like regions, maintain their identity, and drive epithelial cell differentiation

Although mature BALOs consist of epithelial and mesenchymal cell subsets, the model still lacked tissue-resident immune cells that would be required to study processes dependent on these cells in lung development, homeostasis, and disease (Wynn & Vannella, 2016). Therefore, to complete the alveolar niche, TR-Mac were introduced into BALO after isolation from bronchoalveolar lavage (BAL) fluid of adult tdTomato-expressing reporter mice. Single-cell TR-Mac suspensions were prepared, and up to 50 TR-Mac were microinjected at the rate of 6 cells/min under visual control into central regions of day 14 BALO (Appendix Fig S4A and B). Prior to microinjection, the TR-Mac surface antigen signature (CD45+ Ly6g−Gr1−CD11c+Siglec-F+) was confirmed by FACS (Fig 4A). Notably, TR-Mac efficiently engrafted at least until day 28 of culture (Fig 4B). Quantification of the numbers of TR-Mac over time revealed that > 80% of initial TR-Mac could be detected at 10 days post-injection with ~ 87% viability (Appendix Fig S4C and D). Moreover, preservation of the TR-Mac phenotype within the alveolar niche was proven by positive staining of the alveolar macrophage surface markers, CD206 and Siglec-F, 14 days after microinjection (Fig 4C). To demonstrate direct TR-Mac-AEC interaction in the alveolar-like niche of BALO, we performed electron microscopy analyses and revealed TR-Mac filopodia in direct contact with AEC (Fig 4D). Of note, evidence of surfactant uptake and digestion by microinjected TR-Mac was observed in the lumen of the BALO alveolar-like structures (Appendix Fig S4E). A previous publication revealed that TR-Mac express the tight junction molecule connexin 43 (Cx43) upon interaction with AEC in the lung (Westphalen *et al*, 2014). To confirm that TR-Mac would similarly communicate with AEC in BALO after alveolar engraftment, and to visualize this intercellular communication, we stained TR-Mac-supplemented BALO cultures for Cx43 and revealed that TR-Mac within alveolar-like regions, but not TR-Mac mono-cultured in

Matrigel without BALO, expressed Cx43 (Fig 4E). To identify whether TR-Mac impacted epithelial growth and differentiation within BALO, the composition and differentiation stage of the BALO epithelium were evaluated by scRNA-Seq 9 days post-microinjection. Comparative analysis of digested day 23 BALO cultures with and without microinjected TR-Mac showed the presence of 6 distinct clusters defined as club/secretory cells (C1; *Scgb3a2, Muc5b, Bpifa1*), basal cells (C2; *Krt5, Krt14, Aqp3, Trp63*), rMC (C3; *Col1a2, Igfbp4, Apoe*), AEC II (C4; *Sftpc, Cxcl15, Sftpb*), AEC I (C5; *Hopx, Ager, Cldn18*), and ciliated cells (C6; *Foxj1, Tppp3, Lrrc23*) (Fig 4F and Appendix Fig S4F). Data revealed that genes associated with cell proliferation such as *Fos, Fosb, Areg*, and *Klf4* and with inflammatory processes and cellular stress such as *Erg1* and *Atf3* were downregulated in BALO with TR-Mac, whereas genes related to cell differentiation, *Neat1*, and club cell maturation, *Cyp2f2* and *Ces1d*, were upregulated in the cluster containing club/secretory cells (C1) (Fig 4G). The total percentage of terminally differentiated epithelial cells (AEC I and ciliated cells) was significantly higher in BALOs microinjected with TR-Mac (Fig 4H), suggesting that the presence of TR-Mac in BALO drives epithelial differentiation, while rather diminishing cell proliferation and stress signals, thereby accelerating maturation of BALO. In sum, BALOs can be complemented by TR-Mac that engraft into the BALO and exert defined functions that are relevant for lung homeostasis.

## Manipulation of WNT signaling pathway during BALO morphogenesis by knockdown of miR-142-3p gene expression recapitulates developmental defects observed in embryonic lung explants

Following the detailed characterization of the structural and cellular composition of the BALO system, we next tested whether knockdown of important regulatory genes in BALO recapitulated developmental lung defects observed during *in vivo* loss-of-function

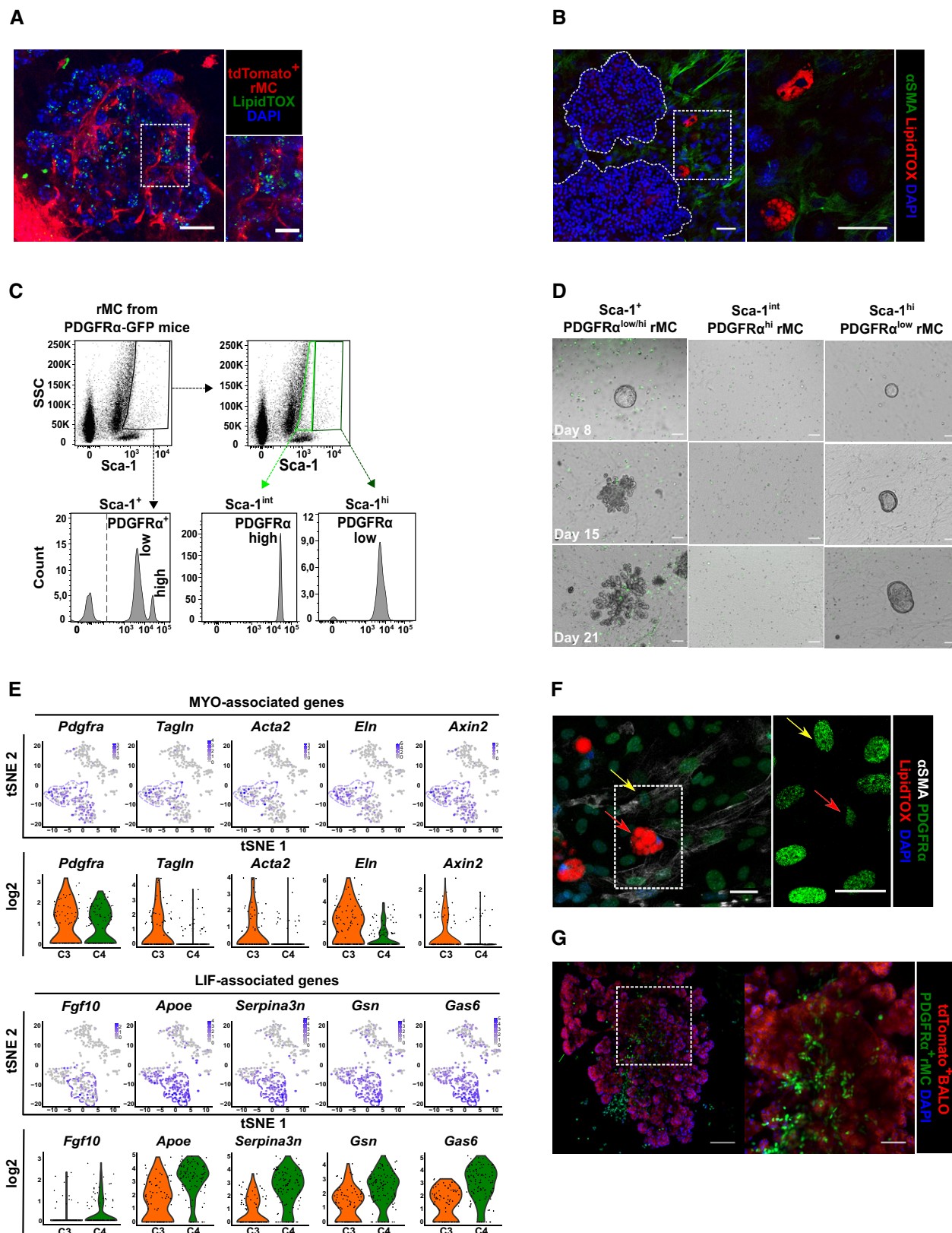

**Figure 3.**

**Figure 3.   Distinct subsets of EpCAM⁻Sca-1⁺ rMC-derived fibroblasts are indispensable for BALO growth, cell differentiation, and branching morphogenesis, and model the mesenchymal niche of the lung.**

A   Representative images of day 21 BALO and tdTomato⁺ rMC stained with LipidTOX (green). Scale bar represents 100 μm.

B   Fluorescence images of αSMA (green) and neutral lipids (LipidTOX red) staining in WT-derived rMC and BALO (dotted lines indicate single BALO or insert) at day 21 of culture. Scale bars represent 50 μm (left) and 25 μm (right).

C   Representative flow cytometric dot plots and histograms of PDGFRα expression in rMC (EpCAM⁻CD45⁻CD31⁻Sca-1⁺) isolated from the lung homogenate of *Pdgfra*^GFP reporter mice.

D   Representative images of BALO formation at day 8, 15, and 21 of co-culture. WT BASCs were co-cultivated with sorted rMC (total PDGFRα+ population) or rMC expressing either low or high levels of PDGFRα-GFP. Scale bars represent 100 μm.

E   tSNE plots and violin plots depicting selected genes representing MYO (*Pdgfrα, Tagln, Acta2, Eln,* and *Axin2*) (top panels) and LIF-associated genes (*Fgf10, Apoe, Serpina3n, Gsn,* and *Gas6*) (bottom panels). Each violin plot shows the frequency distribution of the mean transcript level (log₂). C3 (MYO, orange) and C4 (LIF, green) refer to the scRNA-Seq experiment in Fig 1G.

F   Fluorescence image of αSMA⁺PDGFRα^high MYO (yellow arrows) and LipidTOX⁺PDGFRα^low LIF (red arrows) from sorted PDGFRα-GFP rMC after 21 days of BALO culture. Scale bars represent 50 μm (left) and 25 μm (right).

G   Fluorescence image containing PDGFRα^high MYO from sorted PDGFRα-GFP rMC within BALO derived from tdTomato⁺ mice at day 21 of culture. Scale bars represent 100 μm (left) and 50 μm (right).

experiments. Carraro *et al* (2014) have previously demonstrated that miRNA 142-3p (*miR-142-3p*) controls WNT-dependent mesenchymal progenitor cell proliferation in the developing mouse lung. Inhibition of *miR-142-3p* activity decreases proliferation of the lung mesenchyme and reduces lung growth and branching in embryonic lung explants (Fig 5A). To investigate whether, and if so, to what extent the effect of *miR-142-3p* on the developing embryonic lung was recapitulated in the BALO model, *miR-142-3p*-specific morpholino antisense oligonucleotides (mo142-3p) were diluted in media containing 0.2% FCS and applied to BALO cultures at day 6. Repetitive addition of mo142-3p over a 5-day period at an early organoid stage led to a significant decrease in organoid growth compared with addition of scrambled morpholino control (Scra) without affecting its colony-forming efficiency (Fig 5B and Appendix Fig S5A). Of note, the treatment with mo142-3p did not have adverse effects on cell viability when compared to untreated controls (Appendix Fig S5B). *miR-142-3p* is known to regulate canonical WNT signaling by targeting adenomatosis polyposis coli (*Apc*), a gene involved in the β-catenin destruction complex. Accordingly, gene expression of the *miR-142-3p* target gene *Apc* was significantly upregulated in mo142-3p-treated cultures (Appendix Fig S5C). To identify the cellular compartment where downregulation of WNT signaling occurs within the BALO, TOPGAL mice allowing visualization of active canonical WNT signaling (by monitoring β-galactosidase activity) were employed. For this purpose, BASCs from TOPGAL mice were co-cultured with WT rMC or *vice versa* and treated with mo142-3p or Scra starting at day 6 of culture. Similar levels of β-galactosidase activity were detected in

organospheres before addition of mo142-3p or Scra at day 6 (Fig 5C). Interestingly, treatment with mo142-3p led to a marked decrease in β-galactosidase activity in the epithelial compartment of BALO and in reduced size compared with Scra controls. Visualization of WNT signaling in rMC revealed that mo142-3p also targets the mesenchymal BALO compartment. Of note, β-galactosidase⁺ mesenchymal cells were mainly detected inside the BALO by day 11 of culture (Fig 5D). Our data indicate that WNT signaling is activated in both epithelial and mesenchymal cells during organoid generation. mo142-3p-treated organoids were not only significantly smaller than Scra-treated controls, but also exhibited impaired secondary branching (Fig 5E). Notably, numbers of AEC II and club cells were significantly reduced after treatment with mo142-3p by FACS (Appendix Fig S5D). Knockdown of *miR-142-3p* expression by mo142-3p treatment in flow-sorted EpCAM⁺ epithelial and EpCAM⁻Sca-1⁺ mesenchymal cells from BALO was confirmed by qPCR (Fig 5F). The data demonstrate that (i) WNT signaling is involved in BALO morphogenesis and that (ii) BALO can serve as a platform for genetic manipulation of key developmental pathways to study their involvement in morphogenesis and response to injury.

## BALOs support influenza virus infection and allow modeling of lung infection and injury

Lastly, the applicability of the BALO system for disease modeling was assessed in the context of influenza virus infection. In this regard, we next analyzed whether BALOs support influenza A virus infection using H7N7 and H1N1 IAV strains. To directly visualize infected epithelial cells, recently generated influenza reporter viruses expressing Cre recombinase or GFP, SC35M_{NS1_2A_Cre_2A_NEP} and SC35M_{NS1_2A_GFP-NEP} (SC35M-Cre and SC35M-GFP, H7N7), were used (Reuther *et al*, 2015). To model proximal-to-distal epithelial infection as occurring *in vivo*, BALOs were infected with SC35M-GFP virus by microinjection of the virus suspension into BALO central airway-like structures (Fig 6A). Direct inoculation resulted in viral infection that spread efficiently from the proximal bronchiolar-like regions toward the respective distal alveolar-like regions within 12 h post-infection (pi). Release of infectious influenza A virions from BALO was confirmed by plaque assay at 48 h pi (Fig 6B). To further demonstrate successful BALO infection, gene expression of H1N1 viral nucleoprotein (*Np*) in flow-sorted infected

**Table 1.   Adjusted *P*-value of genes in LIF and MYO clusters.**

| MYO | P_val_adj | LIF | P_val_adj |
|---|---|---|---|
| *Pdgfra* | 3.78E-04 | *Apoe* | 2.26E-26 |
| *Tagln* | 4.37E-07 | *Serpina3n* | 2.00E-30 |
| *Acta2* | 1.02E-06 | *Gsn* | 3.88E-19 |
| *Eln* | 3.44E-15 | *Gas6* | 8.32E-21 |
| *Axin2* | 2.95E-04 | *Fgf10* | 0.00217 |

Adjusted *P*-value of genes associated with MYO (C3) and LIF (C4) phenotypes including *Pdgfrα, Tagln, Acta2, Eln, Axin2* and *Apoe, Serpina3n, Gsn Gas6,* and *Fgf10,* respectively.

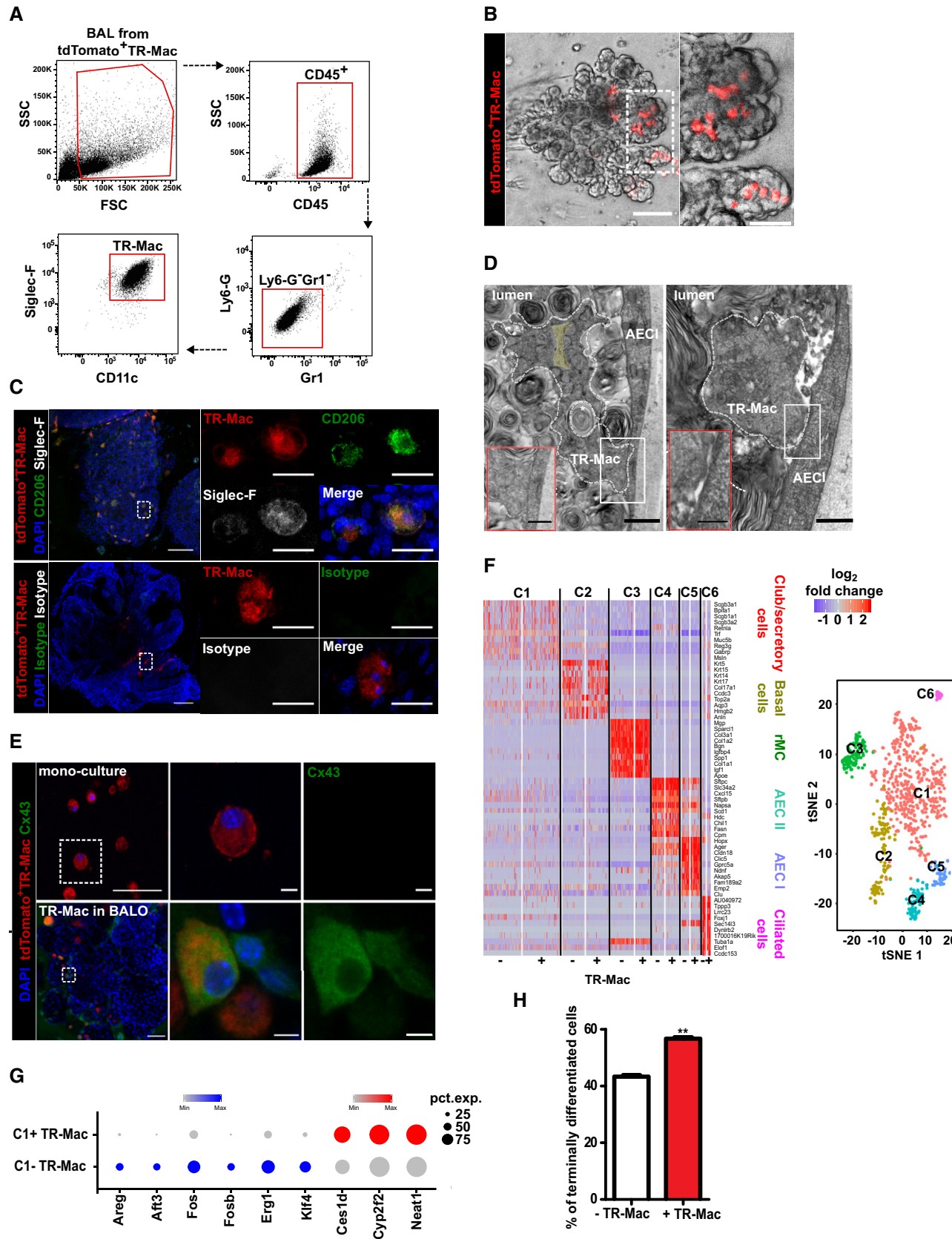

Figure 4.

**Figure 4. TR-Mac engraft into BALO alveolar-like regions, maintain their identity, and drive epithelial cell differentiation.**

A  Gating strategy to define TR-Mac from BAL of adult tdTomato$^+$ mice.

B  Representative images of BALO after microinjection of tdTomato$^+$ TR-Mac at day 14. TR-Mac are preferentially found in alveoli (right). Scale bars represent 100 μm (left) and 50 μm (right).

C  Fluorescence confocal images of CD206, Siglec-F, and isotype control staining 14 days after tdTomato$^+$ TR-Mac microinjection in a day 28 BALO. Scale bars represent 100 μm (left) and 25 μm (right).

D  Electron microscopy depicting filopodium of TR-Mac (white dashed lines) with a characteristic actin filament bundle (yellow background) (left panel) in contact with AEC I within BALO alveolar-like structures. Uptake of surfactant by TR-Mac is depicted by a (*). Scale bar indicates 1,000 nm (left, in insert: 500 nm) and 500 nm (right, in insert: 250 nm).

E  Representative confocal images of Cx43 staining in tdTomato$^+$ TR-Mac monoculture in Matrigel and tdTomato$^+$ TR-Mac microinjected at day 14 and analyzed in a mature day 21 BALO. Scale bars represent 50 μm (overview) and 5 μm (close up).

F  Heat map (left) and tSNE plot (right) of the comparative analysis of digested day 23 BALO cultures with (+) and without (−) microinjected TR-Mac depicting six distinct clusters (C1, club/secretory cells, red; C2, basal cells, yellow; C3, rMC, green; C4, AEC II, blue; C5, AEC I, purple; and C6, ciliated cells, pink).

G  Expression data dot plots of genes found differentially regulated in cluster C1 between day 23 BALO with (C1+) and without (C1−) microinjected TR-Mac. The circle size illustrates the number of cells expressing a specific gene.

H  Percentage of terminally differentiated cells (AEC I and ciliated cells) in day 23 BALO cultures with (+) and without (−) microinjected TR-Mac.

Data information: Bar and dot charts presented as the mean ± SEM and probability determined using $t$-test (**$P$ < 0.01).

EpCAM$^+$/viral hemagglutinin (HA)$^+$ versus non-infected (EpCAM$^+$HA$^-$) cells was analyzed by qPCR. $Np$ gene expression was detected in HA$^+$ but not in HA$^-$ BALO epithelial cells (Fig 6C). Additionally, the proportion of infected EpCAM$^+$ epithelial cells was determined by flow cytometry analysis of intracellular influenza virus NP expression. Viral infection was detected in approximately 8% of the cells (Fig 6D). Identification of IAV-infected cells at a single-cell resolution was achieved by using SC35M-Cre in combination with BALO derived from reporter mice expressing tdTomato that switch from membrane-targeted tdTomato to GFP after Cre-mediated recombination. BALOs were monitored for 25 h pi to score the extent of infection, viral spread, and virus-induced cell loss. We observed viral spread between 10 and 26 h pi, indicating that BALOs fully support viral replication (Fig 6E), which was also visualized over a time course of 26 h using live-cell imaging of $mTmG$ BALO where infection spreads to adjacent cells indicated by color switch from tdTomato to GFP (Movie EV2). Furthermore, 25 h pi we observed significant cell death in alveolar-like areas that were previously infected at 14 h pi (Fig 6E; Herold et al, 2015). To define the host response to infection of BALO, interferon-beta ($Ifnb$) expression in non- and IAV-infected BALO cells was analyzed by qPCR (Fig 6F). A significant upregulation of $Ifnb$ was observed in IAV-infected EpCAM$^+$HA$^+$ BALO cells. To model macrophage–epithelial crosstalk under infection conditions, TR-Mac were microinjected into BALO followed by infection with IAV or control inoculum (mock) for 48 h. Increased release of the pro-inflammatory cytokines TNF-α and IL-6 was detected after BALO IAV infection. Remarkably, TNF-α, IL-6, and IL-1β release was increased in IAV-infected BALO with TR-Mac compared to infected BALO without TR-Mac (Fig 6G). These data indicate that the BALO epithelium not only supports IAV infection and spread but also mounts an antiviral response against IAV infection, which is amplified by TR-Mac.

## Discussion

Several cellular and molecular processes cannot be easily studied *in vivo,* and traditional 2D *in vitro* cultures lack the conditions necessary to study distinct features involved in organ formation and regeneration. Consequently, 3D culture systems have recently emerged as a valuable platform for the better understanding of organogenesis and disease processes *in vitro* (Huch & Koo, 2015). BASCs have been shown to give rise to both airway and alveolar cell lineages upon different types of injury and although BASCs have not yet been identified in the human lungs and our understanding of the biology of these cells is limited, further genome profiling combined with lineage tracing may reveal specific cell markers for BASC identification in mice that could facilitate the discovery of a BASC equivalent in the human lung (Salwig *et al*, 2019). Therefore, in this work we describe the establishment of a robust protocol for isolation and 3D culture of murine progenitor cells with BASC signature and rMC giving rise to complete BALO that exhibits distinct bronchiolar-like and alveolar-like lung structures after 21 days of culture. Our data indicate that a high BALO purity of > 80% can be achieved when enriching these cells in the EpCAM$^{high}$CD24$^{low}$Sca-1$^+$ fraction of the lung homogenate, thus preferentially excluding alveolosphere- and bronchiolosphere-forming progenitor cells. As compared to previously described models, this system models much more closely the structural complexity and epithelial and mesenchymal cell composition of the murine lung with proximal structures containing airway epithelial cells (basal, ciliated, and club/secretory cells) and with distal alveolar-like structures (AEC I and AEC II) including LIF and MYO that directly interact with the epithelium in a spatially defined manner (Barkauskas *et al*, 2013; Lee *et al*, 2014).

Most of the established human lung development and disease models are based on iPSC differentiated in 2D culture, therefore lacking a robust 3D structure including branched airways and distal alveoli (Rock *et al*, 2009; Kaisani *et al*, 2014; Dye *et al*, 2016; Nadkarni *et al*, 2016). Recently, iPSC-derived human lung organoids were demonstrated to form airway-like structures, but maintain a relatively high number of undifferentiated epithelial progenitor cells and low numbers of differentiated bronchial and alveolar cells, particularly AEC I (Dye *et al*, 2015). In addition, generation of lung-like structures from human pluripotent stem cells is relatively time-consuming and requires several months until the final differentiation stages are reached (Chen *et al*, 2017). Nonetheless, in a recent publication by Sachs *et al* (2019), human pseudostratified airway organoids were generated from adult tissue in conditions that allowed long-term expansion of epithelial cells *in vitro*. Those

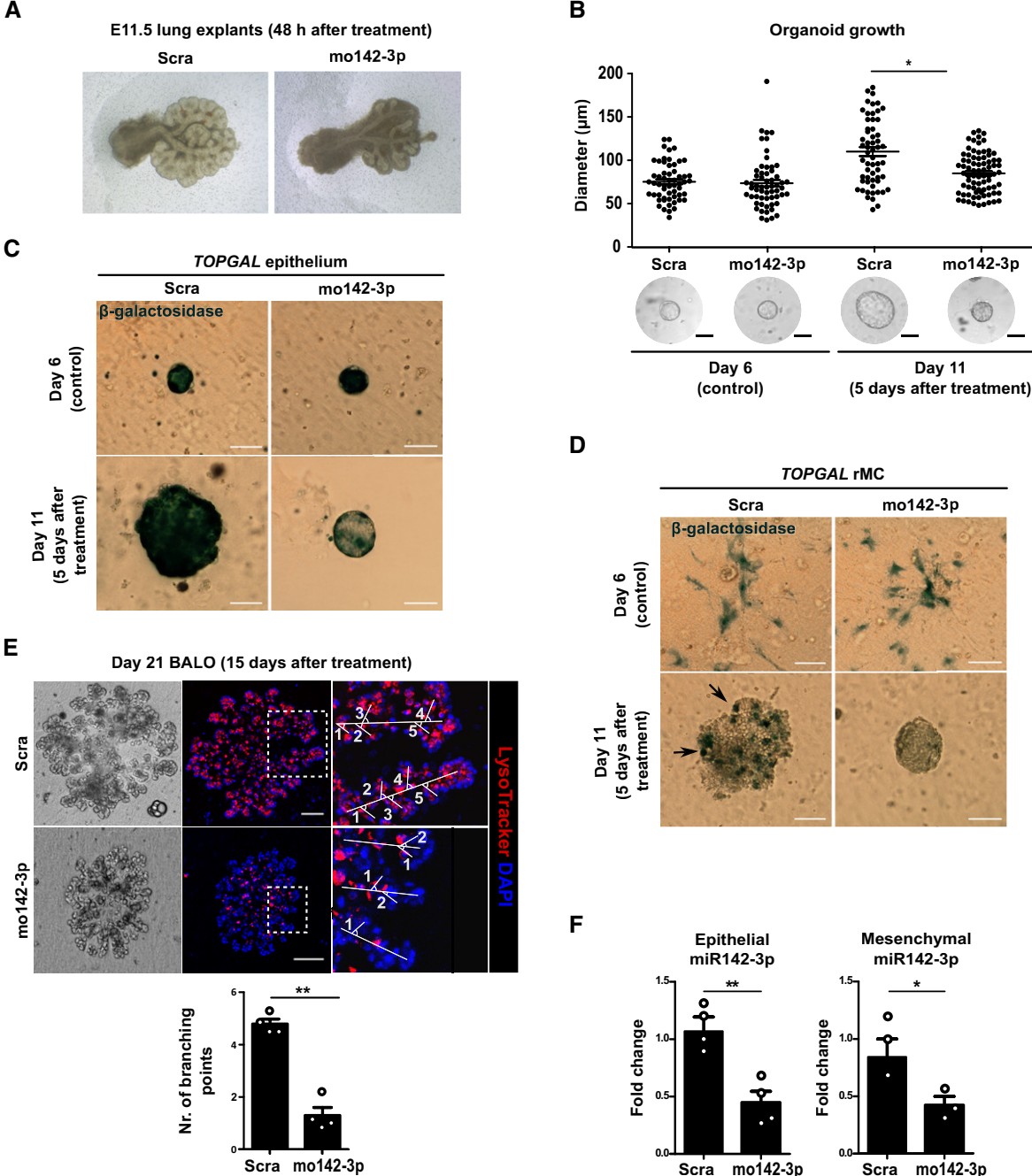

**Figure 5. Manipulation of WNT signaling pathway during BALO morphogenesis by knockdown of miR-142-3p gene expression recapitulates developmental defects observed in embryonic lung explants.**

A Representative images of E11.5 lung explants after treatment for 48 h with 100 μM Scra or mo142-3p reveal reduced size and branching morphogenesis after miR-142-3p knockdown.

B Organoid diameter (*n* = 30–50 per group) in μm before addition of 4 μM Scra or mo142-3p at day 6 (control) and 5 days after treatment (at day 11 BALO culture) in *n* = 3 biological replicates.

C, D Representative images of β-galactosidase staining in *TOPGAL* epithelium (C) and *TOPGAL* rMC (D) BALO cultures before (day 6, control) or 5 days after treatment with either 4 μM Scra or mo142-3p (day 11 of co-culture). BASCs and rMC were isolated from the lung homogenate of TOPGAL mice. β-galactosidase⁺ rMC at day 11 of culture are indicated with arrows.

E Representative transmission and confocal images after LysoTracker staining indicating branching and number of branching points in *n* = 4 BALOs 15 days after treatment with 4 μM Scra or mo142-3p (day 21 of co-culture) in *n* = 3 biological replicates.

F mRNA levels of epithelial and mesenchymal *miR-142-3p* expression in Scra and mo142-3p-treated organoids 5 days after treatment (*n* = 3–4 biological replicates with pooled cells from 4 cultures per replicate).

Data information: Scale bars = 100 μm. Bar charts presented as the mean ± SEM and probability determined using *t*-test (*P < 0.05, **P < 0.01).

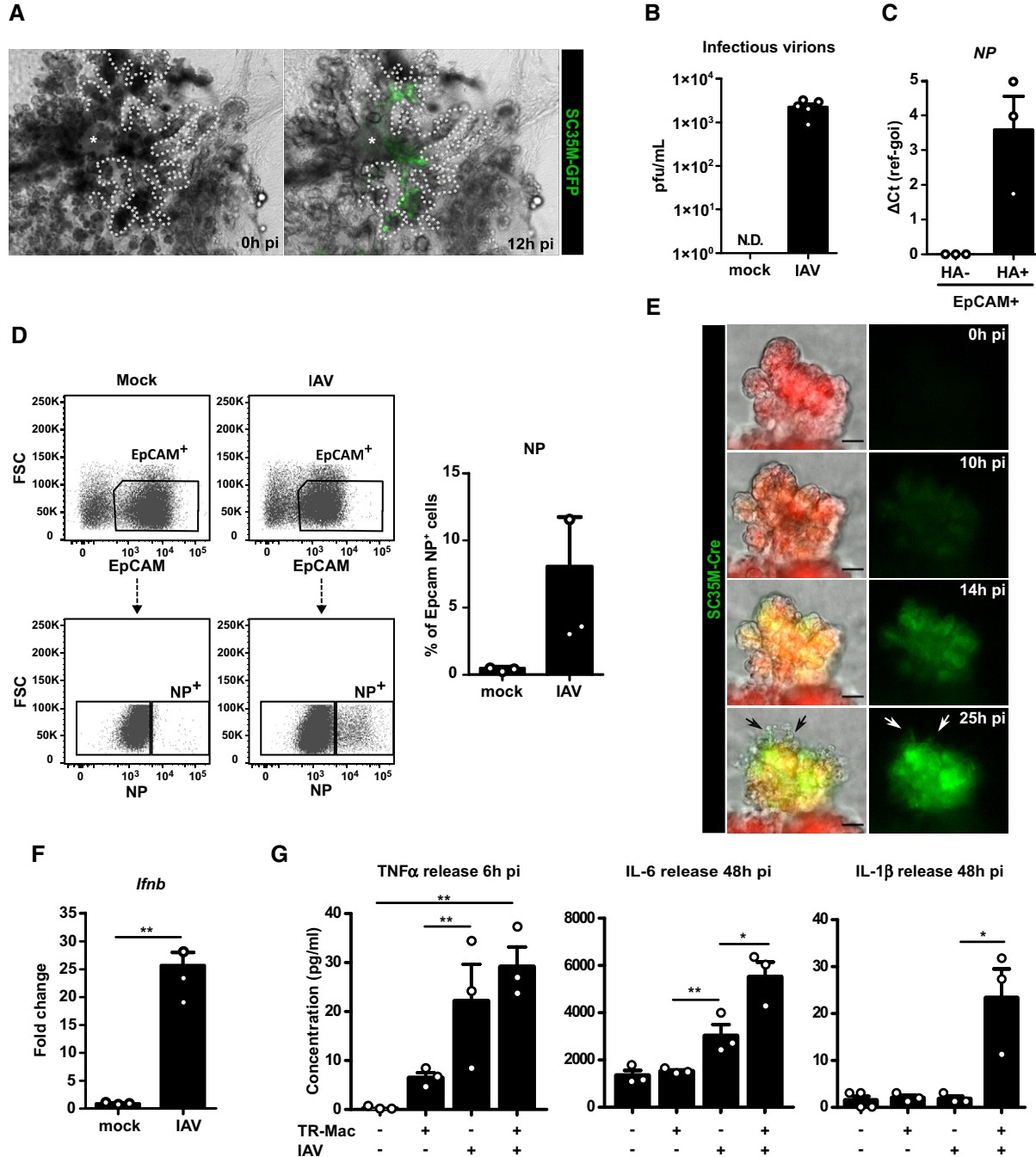

**Figure 6. BALOs support influenza virus infection and allow modeling of lung infection and injury.**

A   Representative images of day 21 BALO after microinjection of SC35M-GFP IAV (green) into the central airway-like (*) after 0 and 12 h pi visualize IVA spread to the alveolar-like regions. Dotted lines illustrate BALO borders. Scale bars represent 100 μm.

B   Quantification of plaque-forming units in supernatants from mock or SC35M IAV-infected BALO 48 h after infection (n = 5 biological replicates). N.D.: not detectable.

C   Relative *NP* expression in PR8 IAV-infected (HA⁺) or non-infected (HA⁻) epithelial cells isolated from BALOs 48 h pi, FACS-sorted according to EpCAM and HA expression (n = 3 biological replicates with pooled cells from 4 cultures per replicate).

D   Representative flow cytometric data showing the percentage of EpCAM⁺NP⁺ cells in mock- and SC35M IAV-infected BALOs at 48 h pi in n = 3 biological replicates with pooled cells from 4 cultures per replicate.

E   Representative fluorescence images of a day 21 distal region in BALO generated from mTmG reporter mouse and infected by SC35M-Cre IAV after 0, 10, 14, and 25 h. Arrows indicate cell death. Scale bars represent 25 and 10 μm in the insert.

F   mRNA expression of *Ifnb* in mock and PR8 IAV-infected BALOs at 48 h pi (n = 3 biological replicates with pooled cells from 4 cultures per replicate).

G   Release of TNF-α (6 h pi), IL-6 (48 h pi), and IL-1β (48 h pi) detected by Bio-Plex® Multiplex immunoassay in the supernatant of mock and IAV-infected BALO cultures with and without TR-Mac (n = 3 biological replicates).

Data information: Bar charts presented as the mean ± SEM and probability determined using *t*-test (F) and one-way ANOVA (G) (*P < 0.05, **P < 0.01).

airway organoids were comprised of basal cells, ciliated cells, and club/secretory cells, thus permitting effective modeling of airway diseases *in vitro* (Sachs *et al*, 2019).

Through investigation of the cellular composition of BALO by scRNA-Seq, qPCR, immunofluorescence, electron microscopy, and FACS, we provide evidence that major cell types of the conducting airways such as basal, secretory, and ciliated cells, as well as alveoli, AEC I and AEC II, are present. Notably, differentiated AEC II within the distal BALO compartment were capable of surfactant secretion. Accordingly, scRNA-Seq revealed expression of early AEC II markers such as *Muc1, Slc34a2,* and *Scd1,* which is indicative of AEC II progressive maturation during BALO formation (Treutlein *et al*, 2014). Furthermore, airway-like regions in BALO at a later stage developed a pseudostratified epithelium and expressed well-known basal cell markers *Aqp3, Krt14,* and *Krt5* indicating the presence of basal-like cells in BALO. To our knowledge, the BALO model is unique in terms of high cellular and structural complexity. Of note, BASC differentiation into basal cells has not been shown *in vivo* (Salwig *et al*, 2019), suggesting that BASC differentiation into basal cells in BALO may be for example driven by the complete absence of basal cells in the BALO airway-like niche.

The phenotype and spatial organization of co-cultured mesenchymal cells emerged as another important aspect of a successful lung organoid system. In this regard, besides providing a complex structural and cellular system, our *in vitro* model allows visualization of epithelial and mesenchymal interactions during BALO formation and differentiation, a feature that is not possible to achieve using current lung air–liquid interface cultures and most organoid systems (Rock *et al*, 2009; Schamberger *et al*, 2015; Rayner *et al*, 2019). During development, mesenchyme-derived growth factors drive epithelial proliferation, differentiation, and branching morphogenesis (Herriges & Morrisey, 2014). Accordingly, CD45$^-$CD31$^-$EpCAM$^-$Sca-1$^+$ rMC proved to be indispensable for proper BALO formation. Previous studies revealed also that LIFs are located at the base of the alveoli and express low levels of PDGFRα, whereas non-lipid-containing MYOs are PDGFRα$^{high}$ and reside at the alveolar entry ring (McGowan & Torday, 1997; Chen *et al*, 2012b). Using rMC isolated from PDGFRα reporter mice, we demonstrate that a PDGFRα-positive fraction of sorted rMC likely contains progenitor cells for LIF and MYO phenotypes. A previous study in which AEC II were co-cultured with PDGFRα$^+$ cells revealed that these cells are necessary for alveolosphere formation (Chen *et al*, 2012b). Accordingly, αSMA$^+$PDGFRα$^{high}$ MYOs were detected inside the organoid structures and seemed to provide a scaffold for tube formation (Barkauskas *et al*, 2013). Nonetheless, BASCs cultured with MYO precursors alone did not show any early organoid formation. In contrast, rMC-derived LipidTOX$^+$PDGFRα$^{low}$ LIF supported BASC proliferation in the early organoid but did not support branching morphogenesis. These data indicate that both PDGFRα$^{high}$ and PDGFRα$^{low}$ mesenchymal cell subsets are indispensable components of the stem cell niche and fulfill distinct tasks during BALO formation in the absence of endothelial cells (Lee *et al*, 2014). Of note, although MYO and LIF phenotype interconversion has been previously reported *in vivo*, it has only been observed under pathological conditions, suggesting that these processes may occur upon injury but not under homeostatic conditions (El Agha *et al*, 2017; Kheirollahi *et al*, 2019). Recently, Zepp *et al* (2017) described a subpopulation of mesenchymal cells in the alveolar

niche co-expressing *Axin2* and *Pdgfra* capable of promoting AEC II self-renewal and differentiation *in vitro*. In this regard, scRNA-Seq demonstrated higher expression of *Axin2* and *Pdgfra* in the MYO cluster, suggesting the presence of *Axin2-Pdgfra*$^+$ cells in BALO cultures, whereas *Lrg5* and *Lrg6* expression was not detected at this time point of culture (Lee *et al*, 2017). Emergence of distinct MYO and LIF phenotypes from PDGFRα$^{high}$ and PDGFRα$^{low}$ rMC within the developing BALO will enable researchers to study epithelial-mesenchymal crosstalk mechanisms and the role of these subsets within the niche during organogenesis, injury, and repair (El Agha *et al*, 2017).

Another important component of the lung stem cell niche is TR-Mac. Macrophage precursors seed the organ as early as embryonic day 10.5 in different waves and significantly contribute to lung development, branching morphogenesis, tissue regeneration, and remodeling (Chen *et al*, 2012a). We have established a microinjection protocol that enables us to successfully engraft these cells into BALO compartments to study and visualize such processes. It has been recently suggested that different subsets of macrophages might exert different roles during branching, alveolarization, and repair after injury (Lechner *et al*, 2017). *Vice versa*, it seems likely that the developing or repairing epithelium emanates signals important for programming of macrophages into distinct phenotypes. However, it has been difficult to address the molecular crosstalk between immune cells and lung epithelia and to visualize such processes. Interactions within BALO between TR-Mac and AEC were detected by electron microscopy, suggesting that this model could facilitate future mechanistic studies. Additionally, scRNA-Seq analysis revealed that addition of TR-Mac induces changes in the transcriptional activity and differentiation of the developing epithelial cells. The AP-1 transcription factor complex has been involved in several cellular functions including proliferation (Eferl & Wagner, 2003). Our data show that genes coding for components of the AP-1 transcription factor complex, *Fos* and *Fosb*, as well as one of its target genes, *Areg*, were downregulated in the presence of TR-Mac. Also, *Erg1* and *Atf3* genes, which regulate the expression of pro-inflammatory genes such as *Ccl4, Il6,* and *Il8,* were downregulated (Park *et al*, 2013). Conversely, genes associated with epithelial cell maturation, *Neat1, Cyp2f2,* and *Ces1d* (Zemke *et al*, 2009; Standaert *et al*, 2014), were upregulated in BALO microinjected with TR-Mac, and BALO with TR-Mac contained a higher percentage of terminally differentiated epithelial cells. These data suggest that macrophages interact with the epithelium to drive cell differentiation, alongside with downregulating genes indicating inflammation-related stress responses, a feature that has been previously ascribed to TR-Mac, where these cells revealed an anti-inflammatory phenotype during lung development (Furukawa *et al*, 2017). Notably, the presence of TR-Mac amplified the pro-inflammatory responses upon infection, suggesting that microinjected TR-Mac may sense the signals from the injured epithelium. These data indicate that our system can be used to study macrophage–epithelium crosstalk in the context of viral infection.

Despite the advantages of the BALO model, we, however, need to mention the lack of an endothelial compartment in our model. Endothelial cells co-develop with the lung epithelium, and it would be of great benefit to study endothelial/epithelial crosstalk *in vitro* or to model lung diseases confined to this compartment. Although CD31$^+$ endothelial cells from adult lungs have been found to drive

organoid development from BASC, a proper spatially organized endo-epithelial network did not develop in such studies (Lee *et al*, 2014). The question remains whether better-defined endothelial progenitor cells will be helpful to increase the cellular complexity of the BALO model.

To test whether BALOs are suitable for genetic loss-of-function approaches, we suppressed *miR-142-3p*, a miRNA known to activate the canonical WNT β-catenin pathway, by addition of morpholino oligos. This approach recapitulated developmental defects in BALOs including reduced growth and impaired branching morphogenesis previously observed in embryonic lung explants treated with mo142-3p (Carraro *et al*, 2014). We reason that the knockdown of target genes at different stages of BALO development is a powerful tool to dissect the function of individual genes without damaging the organoid and without the need for organoid dissociation.

Furthermore, BALO can be used to model diseases that affect the bronchoalveolar compartment of the lung, such as viral infection. We demonstrate that the BALO epithelium infected with different strains of IAV supports viral replication and spread, and mounts an antiviral immune response. We were able to observe IAV infection of epithelial cells in real time using live-cell imaging over a period of 25 h with a recombinant SC35M-Cre reporter virus in combination with BALOs derived from a Cre reporter mouse strain. Microinjection of IAV into BALO airway-like structures recapitulated proximal-to-distal spread of the infection and revealed substantial loss of AEC in the infected alveolar-like areas, modeling this particular aspect of the *in vivo* situation of IAV pneumonia (Herold *et al*, 2015).

Altogether, the BALO model can be applied to (i) visualize lung developmental processes and, prospectively, tissue regeneration after injury induced by infection or other insults at high resolution and over time by microscopic analysis, and (ii) pinpoint the role of cells of mesenchymal, epithelial, and myeloid origin in these processes. A particular advantage of the BALO model is the presence of differentiated mesenchymal cell subsets that interact with the epithelium in a spatially defined manner with distinct roles in epithelial development, recapitulating features of myo- and lipofibroblasts that have been identified *in vivo*. The possibility to supplement the airway/alveolar space of BALO with different tissue-resident or bone marrow-derived leukocyte populations of interest adds a further level of complexity that has not been achieved in previously described models, and will allow visualizing the interactions with epithelial cells of different phenotypes during development, host defense, and repair. Furthermore, the employment of BASCs, rMC, and/or leukocytes carrying gain- or loss-of-function mutations, or from reporter mice, will enable characterization of cell-specific functions of individual genes, a feature that is sometimes difficult to achieve *in vivo* due to the lack of appropriate cell subset-specific transgenic mouse lines. In summary, the BALO model represents a valuable addition to the emerging repertoire of *in vitro* (and *ex vivo*) tools to study pulmonary development and disease.

# Materials and Methods

## Reagents and Tools table

| Reagent/Resource | Reference or Source | Identifier or Catalog Number |
|---|---|---|
| **Experimental models** | | |
| WT (*Mus musculus*) | Charles River Laboratory | *C57BL/6JCrl* |
| mTmG (*M. musculus*) | Jackson Laboratory | *B6.129 (Cg)-Gt (ROSA)26Sor$^{tm4\ (ACTB-tdTomato-EGFP)Luo}$/J* |
| PDGFRα$^{EGFP}$ (*M. musculus*) | Jackson Laboratory | *B6.129S4-Pdgfra$^{tm11(EGFP)Sor}$/J* |
| PDPN$^{EGFP}$ (*M. musculus*) | Jackson Laboratory | *B6;D2-Tg$^{(Pdpn,-EGFP)16Dobb}$/J* |
| UBI-GFP (*M. musculus*) | Jackson Laboratory | *(C57BL/6-Tg$^{(UBC-GFP)30Scha}$/J* |
| *Scgb1a1$^{mCherry}$Sftpc$^{YFP}$* double transgenic mice (*M. musculus*) | Salwig *et al* (2019) | SPC$^{-2A-YFP-2A-tTA-N}$, CCSP$^{-2A-mCherry-2A-tTA-C}$ |
| BASC viewer (*M. musculus*) | Salwig *et al* (2019) | SPC$^{-2A-YFP-2A-tTA-N}$, CCSP$^{-2A-mCherry-2A-tTA-C}$, tetO$_{bi}$$^{lacZ/huGFP}$ |
| BASC v-race (*M. musculus*) | Salwig *et al* (2019) | SPC$^{-2A-YFP-2A-tTA-N}$, CCSP$^{-2A-mCherry-2A-tTA-C}$, tetO$^{biluc/Cre}$, Rosa26$^{stopflox-lacZ}$ |
| **Antibodies** | | |
| Anti-goat Alexa Fluor® 488 | Thermo Scientific | Cat #A-21467 |
| Anti-goat Alexa Fluor® 647 | Thermo Scientific | Cat #A-21469 |
| Anti-mouse Alexa Fluor® 488 | Thermo Scientific | Cat #A-21206 |
| Anti-rabbit Alexa Fluor® 555 | Thermo Scientific | Cat #A-31572 |
| Biotinylated rat anti-mouse CD16/32 | BD Bioscience | Clone 2.4G2 |
| Biotinylated rat anti-mouse CD31 | BD Bioscience | Clone MEC 13.3 |
| Biotinylated rat anti-mouse CD45 | BD Bioscience | Clone 30-F11 |
| Goat influenza A virus HA | Abcam | Cat# ab20841 |
| Goat SCGB1A1 | Santa Cruz Biotechnology | Clone T-18 |

**Reagents and Tools table** (continued)

| Reagent/Resource | Reference or Source | Identifier or Catalog Number |
|---|---|---|
| Mouse Connexin 43 | Thermo Scientific | Clone CX-1B1 |
| Mouse IgG1 Ctl | Abcam | Clone CT6 |
| Mouse influenza A virus NP FITC | Abcam | Cat #ab128193 |
| Mouse β-IV tubulin | Abcam | Clone ONS.1A6 |
| Rabbit Mature SPB | Seven Hills | Cat #WRAB-48604 |
| Rabbit pro-SPB | Seven Hills | Cat #WRAB-55522 |
| Rabbit purified pro-SPC | Millipore | Cat #AB3786 |
| Rabbit SPA | Abcam | Cat #ab115791 |
| Rabbit β-actin | Abcam | Cat #ab8227 |
| Rat CD11c FITC | BioLegend | Clone N4189 |
| Rat CD206 Alexa Fluor® 488 | BioLegend | Clone MMR |
| Rat CD24 PE-Cy7 | BioLegend | Clone M1/69 |
| Rat CD31 Alexa Fluor® 488, PE or Pacific Blue | BioLegend | Clone 390 |
| Rat CD326 (EpCAM) APC-Cy7 or APC-eF780 | BioLegend | Clone G8.8 |
| Rat CD45 FITC, APC-C7 or Horizon™ V450 | BD Biosciences | Clone 30-F11 |
| Rat CD45 V500 | BioLegend | Clone 30-F11 |
| Rat GR1 PE-Cy7 | BioLegend | Clone RB6-8C5 |
| Rat IgG2a Ctl | BioLegend | Clone RTK2758 |
| Rat Ly6A/E (SCA-1) APC or Pacific Blue | BioLegend | Clone D7 |
| Rat Ly6G APC | BioLegend | Clone 1A8 |
| Rat Siglec-F APC-Cy7 | BD Biosciences | Clone E50-2440 |
| Siglec-F eFluor 660 and | Thermo Scientific | Clone 1RNM44N |
| Swine anti-rabbit IgG | Agilent | Cat #F0054 |
| Syrian hamster IgG Ctl | BioLegend | Clone SHG-1 |
| Syrian hamster PDPN APC | BioLegend | Clone 8.1.1 |
| **Oligonucleotides and other sequence-based reagents** | | |
| 5S rRNA | This study | FP 5′-TCTCGGAAGCTAAGCAGGGTC-3′; RP 5′-AGCCTACAGCACCCGGTATTC-3′ |
| Apc | This study | FP 5′-CATGGACCAGGACAAAAACC-3′ RP5′-GAACACACACAGCAGGACAGA-3′ |
| Foxj1 | This study | FP 5′-GAGCCAGGCCTCACATTCG-3′ RP 5′-CGCTGGTAACCCAGACTCC-3′ |
| Hopx | This study | FP 5′-CAACAAGGTCAACAAGCACCC-3′ RP 5′-GCGCTGCTTAAACCATTTCTGC 3′ |
| Ifnb | This study | FP 5′-GTTACACTGCCTTTGCCATCC-3′ RP 5′-GTGGAGTTCATCCAGGAGACG-3′ |
| miR-142-3p | This study | FP 5′-ACTCCAGCTGGGTGTAGTGTTTC CTACTT-3′; stem loop reverse 5′-CTCAAC TGGTGTCGTGGAGTCGGCAATTCAGTTGAGTCCATAAA-3′ |
| miRNA | This study | Universal reverse 5′ TCAACTGGTG TCGTGGAGTCG-3′ |
| Muc5ac | This study | FP 5′-AGGACCACTGTATTGCTGGC-3′ RP 5′-TCCAGAACATGTGTTGGTGC-3′ |
| NP | This study | FP 5′-ACGAAGGTGGTCCCAAGAGG-3′ RP 5′-GA TTTGGCCCGCAGATGCC-3′ |
| Rps-18 | This study | FP 5′-CCGCCATGTCTCTAGTGATCC-3′ RP 5′-TTGGTGAGGT CGATGTCTGC-3′ |
| Sftpc | This study | FP 5′-GGAGGAAGGGCATGATACTGG-3′ RP 5′-TTCTACCGACCCTGTGGATGC-3′ |
| miR-142-3p-specific morpholino | Gene Tools, LLC | 5′TCCATAAAGTAG GAAACACTACACC-3′ |
| Scramble morpholino | Gene Tools, LLC | 5′-CCTCTTACCTCAGT TACAATTTATA-3′ |

**Reagents and Tools table** (continued)

| Reagent/Resource | Reference or Source | Identifier or Catalog Number |
|---|---|---|
| **Chemicals, enzymes, and other reagents** | | |
| Biotin-binder magnetic beads | Life Technologies | Cat #11947 |
| Insulin-Transferrin-Selenium | Biozym | Cat #41400-045 |
| L-Glutamine | Gibco | Cat #11539876 |
| Matrigel matrix | Corning | Cat #356231 |
| Paraformaldehyde | Merck | Cat #104005 |
| αMEM | Gibco | Cat #41061029 |
| DMEM | Gibco | Cat #12491023 |
| Dispase | Corning | Cat #354235 |
| DNase I | Serva | Cat #18535.01 |
| Heparin | Stem cell Technologies | Cat #07980 |
| **Software** | | |
| DIVA software (v8.02) | https://www.bdbiosciences.com | |
| FlowJo (v6.10.1) | https://www.flowjo.com | |
| GraphPad PRISM 5 (v5.01) | https://www.graphpad.com | |
| ImageJ (v1.53c) | https://imagej.nih.gov/ij | |
| R (v3.3.3) | https://cran.r-project.org/web/packages/Seurat/index.html | |

### Methods and Protocols

#### Mice

Mice were between 8 and 10 weeks of age and were housed under specific pathogen-free conditions with free access to food and water. All animal experiments were approved by the responsible animal ethics committee and by the local authorities.

#### Primary cell isolation

Primary murine lung cell isolation was modified from Herold *et al* (2006).

(i) Mice were anesthetized with isoflurane inhalation and killed by cervical dislocation.

(ii) Lungs were perfused with 20 ml of sterile HBSS via the right ventricle until they were visually free of blood.

(iii) To obtain lung homogenate, 1.5 ml dispase was instilled through the trachea followed by lung harvest and 40-min incubation at room temperature (RT).

(iv) Trachea was removed, and the lungs were homogenized in DMEM/2.5% HEPES with 0.01% DNase I using the gentleMACS® Dissociator (MACS Miltenyi Biotec).

(v) Cell suspension was filtered through 70- and 40-μm nylon filters and centrifuged for 10 min, 500 $g$ at 4°C.

(vi) To remove red blood cells (RBCs), cells were incubated in 1 ml RBC lysis buffer (1.5 M $NH_4Cl$, 100 mM $NaHCO_3$, 10 mM EDTA, pH 7.4) for 90 s.

(vii) Lysis was stopped with addition of DMEM-containing 10% FCS followed by and cell centrifugation for 10 min, 500 $g$ at 4°C.

(viii) For leukocyte/endothelial cell depletion, cells were counted and the Dynabeads Magnetic Separation Technology protocol was followed (Thermo Fischer) to obtain the exact volume of antibody and beads needed.

(ix) Cells were incubated with an antibody mix containing biotinylated rat anti-mouse CD45, CD16/32, and CD31 antibodies for 30 min at 37°C.

(x) Cells were centrifuged for 10 min, 500 $g$ at 4°C, and resuspended in DMEM-containing streptavidin-linked Magne-Sphere Paramagnetic Particles for 30 min with rotation at RT.

(xi) For the magnetic separation, cells were placed on the magnetic stand for 15 min at RT.

(xii) CD45/CD16/32/CD31-negative cells in the flow-through were collected and washed once with DMEM to be used for further analysis.

#### Flow cytometry and cell sorting

Multicolor flow cytometry and cell sorting were performed on a BD LSRFortessa™ and a BD FACSAria™ III cell sorter using DIVA software (BD Bioscience).

(i) Following primary cell isolation, cells were pelleted by centrifugation for 10 min, 500 $g$ at 4°C, and resuspended in MACS buffer (PBS, 7.4% EDTA, 0.5% FCS, pH 7.2).

(ii) Cells were stained with the antibody cocktail-containing gamma-globulins (sandoglobulin) (1:10), CD31 Alexa 488 (1:50), CD45 FITC (1:50), CD326 APC/Cy7 (1:50), CD24 PE/Cy7 (1:200), and Sca-1 PB (1:50), and for 15 min at 4°C in MACS buffer.

(iii) Cells were washed once with MACS buffer and centrifuged for 5 min, 500 $g$ at 4°C; resuspended in 300–500 μl of MACS buffer; and filtered with 40 μm mesh into FACS tubes for cell analysis.

(iv) Cell sorting was performed using an 85-μm nozzle after doublet and dead cell exclusion.

#### BALO culture

Flow-sorted EpCAM$^{high}$Sca-1$^+$CD24$^{low}$ cells and EpCAM$^-$Sca-1$^+$ rMC were co-cultured in growth factor-reduced Matrigel® (Corning) as follows:

(i) Flow-sorted cells were centrifuged for 5 min, 500 $g$ at 4°C, and resuspended in medium containing α-MEM, 10% FCS, 100 U/ml penicillin, 0.1 mg/ml streptomycin, 2 mM L-glutamine, 1 × insulin/transferrin/selenium, and 0.0002% heparin.

(ii) For co-cultivation, cell concentration was adjusted to $5 \times 10^3$ EpCAM$^{high}$CD24$^{low}$Sca-1$^+$ cells and $3 \times 10^4$ rMC per 25 µl of medium per cell culture insert.

(iii) Mix of EpCAM$^{high}$CD24$^{low}$Sca-1$^+$ cells and rMC (50 µl per insert) was diluted with cold growth factor-reduced Matrigel® (Corning) at a 1:1 ratio as described before (Quantius *et al*, 2016).

(iv) For each insert, 90 µl of mixed EpCAM$^{high}$CD24$^{low}$Sca-1$^+$ cells and rMC in Matrigel was added on top of a 12-mm cell culture insert, placed in a 24-well plate, and incubated for 5–10 min at 37°C for polymerization of Matrigel.

(v) To obtain an air–liquid interface, 350 µl of medium was added per well and cultures were incubated at 37°C with 5% $CO_2$ until further analysis.

(vi) Media were changed three times per week.

### Organoid digestion and passaging

For passaging, FACS analyses, and single-cell RNA-Seq, BALOs were removed from cell culture inserts by addition of ice-cold PBS and rigorous pipetting, collected in 1.5-ml tubes, and centrifuged. The pellet was resuspended in 200 µl dispase and incubated for 30 min at 37°C with rotation. 1 ml DMEM was added, and tubes were centrifuged. Cells were washed twice with MACS buffer, stained with antibodies, and analyzed by flow cytometry, or resuspended in media and co-cultured in Matrigel with freshly sorted rMC for passaging, alternatively 2,300 cells/µl in 1 × PBS/0.1% BSA for single-cell RNA-Seq.

### Quantitative real-time PCR

RNA was isolated from flow-sorted cells and BALO cultures using RNeasy Micro Kit or miRNeasy Mini Kit (Qiagen). cDNA was synthesized using M-MLV reverse transcriptase (Invitrogen) and random hexamer primer according to the manufacturer's instructions. For reverse transcription of *miR-142-3p*, a stem loop reverse primer was used according to the protocol published by Tang *et al* (2006). Quantitative PCR was performed in a StepOnePlus™ Real-Time PCR System (Thermo Scientific) using SYBR Green. The relative gene expression levels were normalized to ribosomal protein subunit S18 (RPS-18) or 5S rRNA as indicated and presented as fold change in gene expression relative to day 0, scramble, or mock.

### Single-cell RNA sequencing

Single-cell RNA-Seq was performed as recommended by the manufacturer (Illumina protocol version: 1000000021452 v01). Briefly, cells were mixed with lysis buffer, barcoding bead suspension, and reverse transcription reagent. An emulsion-containing droplets with single cells and a barcode bead were generated with the ddSEQ Single-Cell Isolator (Bio-Rad). Within the droplets, cells were lysed, allowing for binding of mRNA to the beads prior to reverse transcription. cDNA was purified and complemented by second-strand cDNA synthesis. The resulting cDNA was fragmented with tagment enzyme and amplified. Finally, libraries were cleaned up, pooled, and subsequently sequenced on a NextSeq 500 System (Illumina). Upon sequencing, barcodes belonging to cells were identified using

the Illumina BaseSpace platform. Reads were demultiplexed with a custom python script (available on request). Reads were mapped in the mouse genome using STAR (version 2.5.3a). Mapped reads were counted with the software featureCounts. Following, UMI-tools was used to count the unique molecules per gene. Downstream analysis was performed in R (version 3.3.3) with the Seurat package (version 2.2, https://cran.r-project.org/web/packages/Seurat/index.html). Genes expressed in less than three cells were excluded, and cells with less than 200 detected genes were removed. UMI and number of genes ranged between 1,630 and 3,320 per cell. After filtering, the count values for each cell were normalized by using the NormalizeData method from Seurat and variable genes were detected with the FindVariableGenes method. To remove unwanted variation sources within the genes, data were regressed using the ScaleData method from Seurat combined with the number of UMIs per cell. Principal component analysis (PCA) was performed on the data, and results were visualized by using non-linear dimensional reduction (tSNE). Finally, up- and downregulated genes were identified for each cluster and visualized as heat maps and violin plots. For the comparative analysis between BALO cells grown with and without macrophages (two conditions), a canonical correlation analysis (CCA) was performed using Seurat's RunCCA method. The top 1,000 genes with the highest variability for each condition were selected. Like the Seurat analysis of the data sets before, the cells were then clustered with the FindClusters function using a resolution of 0.4. UMI and number of genes ranged between 2,713 and 3,287 per cell. To identify possible detected cell types, marker genes (upregulated genes) conserved across the two conditions for each cluster were analyzed with the FindConservedMarkers function. The results were visualized as heat maps, violin plots, and dot plots using the DoHeatmap, VlnPlot, and SplitDotPlotGG methods.

### Morpholino antisense oligo transfection

Scramble morpholino (Scra) and *miR-142-3p*-specific morpholino (mo142-3p) were prepared in αMEM with 0.2% FCS at a concentration of 4 µM. At day 6 of culture, mo1423p or Scra was applied to BALOs overnight for three consecutive periods of 12 h, followed by application for 48 h for days 9–11. At day 11, treated organoids were removed from Matrigel and flow-sorted based on EpCAM expression. Toxicity testing was performed using propidium iodide staining (1:100 dilution) followed by FACS analysis. For organoid formation experiments, BALOs were treated every 24 h with mo142-3p or Scra until day 21 of culture.

### Electron microscopy

Day 21 BALOs were fixed for 4 h in 1.5% glutaraldehyde (Merck, Darmstadt, Germany) and 2.5% PFA in 0.1 M phosphate buffer. Cultures were then washed in Tris–HCl and osmicated for 2 h at RT in 2% OsO4 (Sigma-Aldrich, St. Louis, USA) diluted in distilled water. Samples were rinsed five times for 3 min in distilled water and stained overnight *en bloc* in half-saturated uranyl acetate (Merck) in distilled water. Lastly, samples were rinsed again five times for 3 min in distilled water and were then routinely embedded in epon resin (Sigma-Aldrich) through a graded series of ethanol and propylene oxide. Ultrathin sections (60–90 nm) were cut with a Reichert Ultracut E ultramicrotome (Leica, Bensheim, Germany) and viewed with a transmission electron microscope (EM 902; Zeiss, Wetzlar, Germany) equipped

with a slow scan 2 K CCD camera (TRS; Tröndle, Moorenweis, Germany). Images for illustration were corrected for overall brightness, but not manipulated otherwise.

### LysoTracker, phospholipid, neutral lipid, and viability staining

For staining of lamellar bodies, LysoTracker® Red DND-99 (Thermo Scientific) stock solution was diluted in pre-warmed α-MEM according to the manufacturer's instructions. 50 nM LysoTracker was added to co-cultures and incubated for 30 min at 37°C. Similarly, for staining of phospholipids, LipidTOX™ red phospholipidosis detection reagent (Thermo Scientific) was diluted 1:500 in α-MEM and added to BALOs for 48 h. Microscopic pictures of co-cultures were taken using the EVOS™ FL Auto Imaging System (Thermo Scientific). For staining of neutral lipids, co-cultures were fixed with 4% PFA for 15 min, washed twice with PBS, and stained with 1× of PBS diluted LipidTOX™ neutral lipid staining solution (Thermo Scientific). After 6-h incubation at RT, cultures were co-stained with DAPI and mounted on glass slides for imaging. For the TR-Mac viability staining, the cell viability imaging kit (Thermo Scientific) was used according to the manufacturer's instructions. A Leica SP8 microscope was employed for confocal imaging.

### Immunofluorescence

BALO cultures were fixed with 4% PFA for 10 min at RT. Cultures were washed with buffer (0.2% BSA/PBS), permeabilized with 0.1% Triton X-100/PBS for 5 min, and blocked in 5% horse serum, 2% BSA/PBS for 30 min at RT. For staining, cultures were incubated with primary antibodies over night at 4°C, followed by washing and addition of secondary antibodies for 2 h at RT. For the staining of PDGFRα-GFP cultures, cultures were left unpermeabilized.

### Western blot

Lysis buffer containing 50 mM Tris–HCl/pH 8.0, 5 mM EDTA, 150 mM NaCl, 1% ($v/v$) Triton X-100, 0.5% ($w/v$) Na-deoxycholate, and 1 mM PMSF was used for preparation of the protein extract. The probes were loaded to SDS–PAGE, followed by transfer onto PVDF membranes (Roth) in a wet blotting chamber according to the manufacturer's protocol (Bio-Rad). Obtained immunoblots were blocked by incubating at RT for 1 h in blocking buffer containing either 5% ($w/v$) non-fat milk for SPA or 3% ($w/v$) BSA for other antibodies. Next, immunostaining for the proteins of interest was performed overnight at 4°C. The primary antibodies used for Western blotting were mature SPB (1:1,000), pro-SPB (1:1,000), mature SPC (1:1,000), pro-SPC (Millipore, AB3786 1:1,000), and SPA (1:1,000). The blots were then incubated with horseradish peroxidase-conjugated secondary swine anti-rabbit IgG antibodies (1:10,000) for 1 h at RT. The Immobilon Western Chemiluminescent HRP substrate (Millipore) was used for the blots development, and emitted signal was detected with a chemiluminescence imager (Intas ChemoStar, Intas). Thereafter, blots were stripped for 20 min using Stripping Buffer (0.1 M glycine, pH 2.5), followed by re-probing the blots with antibodies against the loading control β-actin (1:5,000).

### β-galactosidase staining

BALOs were fixed with 4% PFA for 5 min at RT, washed once with PBS, and incubated with pre-warmed LacZ buffer (5 mM [Fe(CN)$_6$]$^{3-}$, 5 mM [Fe(CN)$_6$]$^{4-}$, 2 mM MgCl$_2$ in PBS) for 10 min at 37°C. LacZ buffer was replaced by staining solution (1 mg/ml X-Gal in LacZ buffer) and incubated at 37°C. Stained cultures were rinsed once with PBS and kept in PBS at 4°C for further analysis.

### Infection of BALO

BALOs were washed once in PBS and infected with $5 \times 10^6$ pfu (plaque-forming units) of SC35M IAV (H7N7) or A/PR/8/34 (PR8) (H1N1, mouse-adapted) in 600 μl of ice-cold PBS for 2 h at RT and gentle agitation (200 rpm), followed by 30-min incubation at 37°C with 5% CO$_2$. The inoculum was removed, and inserts were washed twice with PBS. Inserts were placed in infection medium (α-MEM, 0.2% BSA, 100 U/ml penicillin, 0.1 mg/ml streptomycin, 1 × insulin/transferrin/selenium, 2 mM L-glutamine, 0.0002% heparin) and incubated for the indicated times. For direct inoculation approach, culture medium was removed and replaced by medium without additives, followed by microinjection of BALO airway-like structures with SC35M$_{NS1\_2A\_GFP-NEP}$ (SC35M-GFP) IAV and SC35M$_{NS1\_2A\_Cre\_2A\_NEP}$ (SC35M-Cre) IAV, diluted in PBS (Reuther *et al*, 2015). After an hour incubation, media were replaced by infection media and incubated for the indicated times. Images of infected BALOs were taken using EVOS™ FL Auto Imaging System (Thermo Scientific). The data shown are representative of three independent experiments.

### Plaque assay

Infection of MDCK cells was performed in duplicate in 12-well plates, using BALO supernatants at 48 h pi serially diluted in PBS for 1 h at 37°C. Pre-warmed Avicel® overlay medium (MEM, 0.1% BSA, 100 U/ml penicillin, 0.1 mg/ml streptomycin, 0.1% HEPES, 2 μg/ml trypsin (TPCK-treated), 1.25% Avicel) was added. Plates were incubated at 37°C, 5% CO$_2$ without movement. After 48 h, the Avicel overlay was removed, and cells were fixed with 4% PFA in PBS for 15 min at RT. Cells were washed twice with PBS and permeabilized with 0.3% Triton X-100 in PBS for 15 min at RT. Staining for viral protein was done using a polyclonal antibody against IAV NP (Abcam) in staining buffer (PBS, 10% horse serum, 0.1% Tween-80) for 1 h at RT. Cells were washed three times with washing buffer (PBS, 0.05% Tween-80) and incubated with horseradish peroxidase-conjugated secondary antibody (Santa Cruz) followed by further washing steps. NP staining was visualized using TrueBlue™ peroxidase substrate (KPL), and stained plaques were counted.

### Cytokine measurement

Release of the pro-inflammatory cytokines IL-6, TNF-α, and IL-1β was measured from the supernatant of mock- and IAV-infected BALO cultures supplemented with and without TR-Mac employing the Bio-Plex® Multiplex Immunoassay System with Bio-Plex Manager™ software (Bio-Rad) according to the manufacturer's instructions.

### Confocal microscopy, live-cell imaging, and microinjection

For live-cell imaging of BALO formation from day 15–20 of culture, a Leica DMI6000 B microscope was used. A Leica SP8 microscope was employed for confocal imaging. On day 14 of culture, microinjection of TR-Mac isolated from the BALF of mTmG mice into the lung organoids was performed. For BALO infection imaging with SC35M-Cre, the EVOS™ FL Auto Imaging System (Thermo Scientific) was used during the first 48 h. For SC35M-GFP virus

inoculation, viral infection was achieved by direct injection into the airway-like regions within BALO. All microinjections were performed using a 10-μm-diameter capillary needle (Eppendorf). Cultures were kept in the incubator with 5% $CO_2$ at 37°C until further analysis.

### Culture of embryonic lung explants

Timed-pregnant WT mice were sacrificed on post-coitum day 11.5 (E11.5). Embryos were harvested, and lung explants were placed on 8 μm Nuclepore Track-Etch membranes and cultured in DMEM:F12 medium (Gibco) with 0.5% FBS. Mo142-3p and Scra (Gene Tools) were added at 100 μM to the lung explants for 48 h.

### Statistics

All data are given as mean ± SEM. Statistical significance between two groups was estimated using the unpaired two-sided Student's *t*-test. For comparison of three groups, one-way ANOVA and *post hoc* Tukey's test were performed. Calculations were done with GraphPad Prism 5. A *P*-value lower than 0.05 was considered to be significant.

## Data availability

The data sets and computer code produced in this study are available in the following database: Single-cell RNA-Seq data: SRA accession number PRJNA473688 (http://www.ebi.ac.uk/ena/data/view/PRJNA473688).

Additional data are available from authors upon reasonable request.

**Expanded View** for this article is available online.

## Acknowledgements

This study was supported by the German Research Foundation (DFG; SFB 1021 C05 and Z02, SFB-TR84 B9 and A6, KFO 309 P2/P5/P6/P7/P8/Z01; SFB 1160 to MS; Excellence Cluster Cardio Pulmonary System), the University Hospital Giessen and Marburg (FOKOOPV to SH and SB), and the German Center for Lung Research (DZL). SB was supported by DFG (BE4443/4-1; BE4443/6-1; and SFB1213). EE was also funded by SFB CRC1213-project A04 and acknowledges the support of the Institute for Lung Health (ILH), DZL and the Cardio-Pulmonary Institute (CPI, EXC 2026, Project ID: 390649896). We also acknowledge technical assistance by the Bioinformatics Core Facility/Professorship of Systems Biology at JLU Giessen and access to resources financially supported by the BMBF grant FKZ 031A533 to the BiGi center within the de.NBI network. We thank Dr. Katrin Ahlbrecht for kindly providing the PdpnGFP mice. This publication uses data collected within the frame work of the first author's PhD dissertation "Establishment of murine 3D bronchioalveolar lung organoids from adult somatic stem cells for organ development and disease modeling" published in 2018 at the Justus-Liebig-Universität Gießen, Germany. Open access funding enabled and organized by Projekt DEAL.

## Author contributions

AIV-A, MH, EEA, REM, IV, MS, SB, TB, WS, and SH were involved in study design and concept. AIV-A, Mo.He, MCH, AS, GC, JPM, and WK acquired the data. AIV-A, AH, JPM, TH, and AGo were involved in the ScRNA-Seq experimental design and analysis. AIV-A, MH, EEA, ISa, ISh, AGü, and SH conducted other data analysis, interpretation, and statistics.

## Conflict of interest

The authors declare that they have no conflict of interest.

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
