## [Review Process File · The EMBO Journal]

Multilineage murine stem cells generate complex organoids to model distal lung development and disease

Ana Vazquez-Armendariz, Monika Heiner, Elie El-Agha, Isabelle Salwig, Andreas Hoek, Marie Hessler, Irina Shalashova, Amit Shrestha, Gianni Carraro, Jan Mengel, Andreas Guenther, Rory Morty, Istvan Vadasz, Martin Schwemmle, Wolfgang Kummer, Torsten Hain, Alexander Goesmann, Saverio Bellusci, Werner Seeger, Thomas Braun, and Susanne Herold
DOI: doi.org/10.15252/emboj.2019103476

Corresponding author: Susanne Herold (Susanne.Herold@innere.med.uni-giessen.d)

Review Timeline:

Submission Date:	20th Sep 19
Editorial Decision:	4th Nov 19
Revision Received:	29th Jun 20
Editorial Decision:	21st Jul 20
Revision Received:	14th Aug 20
Accepted:	24th Aug 20

Editor: Daniel Klimmeck

Transaction Report:

Dear Dr Herold, dear Dr Vazquez-Armendariz,

Thank you for the submission of your manuscript (EMBOJ-2019-103476) to The EMBO Journal. Your manuscript has been sent to three reviewers, and we have reports from all of them, which I enclose below.

As you will see, the referees acknowledge the potential interest and novelty of your results, although they also express a number of major issues that will have to be addressed before they can support publication of your manuscript in The EMBO Journal. In more detail, referee #1 states that the lineage composition of the generated bronchio-alveolar organoids remains unclear and questions the physiological relevance of your model (ref#1, standifrst pts.1,2). In addition, this referee asks you to add complementary analyses on the macrophage population studied and its effect on the organoids (ref#1, pts.6-8). In line, referee #2 is concerned that the BASC stem cell contribution to basal lineages in your model and their ex vivo colony-forming capacity have not been conclusively demonstrated (ref#2, pts. 1,2). In addition, the reviewers raise a number of issues related to statistical robustness and quantification of your findings as well as improved quality of stainings and other controls required that would need to be conclusively addressed to achieve the level of robustness and clarity needed for The EMBO Journal.

I judge the comments of the referees to be generally reasonable and given their overall interest, we are happy to invite you to revise your manuscript experimentally to address the referees' comments.

Please let me know any time if you have additional questions or need further input on the referee comments.

Please see below for additional instructions for preparing your revised manuscript.

Thank you for the opportunity to consider your work for publication. I look forward to your revision.

Kind regards,

Daniel Klimmeck

Daniel Klimmeck, PhD
Editor
The EMBO Journal

Before submitting your revision, primary datasets (and computer code, where appropriate) produced in this study need to be deposited in an appropriate public database (see

<https://www.embopress.org/page/journal/14602075/authorguide#datadeposition>).

The accession numbers and database should be listed in a formal "Data Availability" section (placed after Materials & Method) that follows the model below (see also

<https://www.embopress.org/page/journal/14602075/authorguide#availabilityofpublishedmaterial>).

Please note that the Data Availability Section is restricted to new primary data that are part of this study.

Data availability

- RNA-Seq data: Gene Expression Omnibus GSE46843

(<https://www.ncbi.nlm.nih.gov/geo/query/acc.cgi?acc=GSE46843>)

- [data type]: [name of the resource] [accession number/identifier/doi] ([URL or identifiers.org/DATABASE:ACCESSION])

Our journal also encourages inclusion of *data citations in the reference list* to directly cite datasets that were re-used and obtained from public databases. Data citations in the article text are distinct from normal bibliographical citations and should directly link to the database records from which the data can be accessed. In the main text, data citations are formatted as follows:

"Data ref: Smith et al, 2001" or "Data ref: NCBI Sequence Read Archive PRJNA342805, 2017". In the Reference list, data citations must be labeled with "[DATASET]". A data reference must provide the database name, accession number/identifiers and a resolvable link to the landing page from which the data can be accessed at the end of the reference. Further instructions are available at <https://www.embopress.org/page/journal/14602075/authorguide#referencesformat>

- a point-by-point response to the referees' comments, with a detailed description of the changes made (as a word file).

- a word file of the manuscript text.

- individual production quality figure files (one file per figure)

- a complete author checklist, which you can download from our author guidelines

(<https://www.embopress.org/page/journal/14602075/authorguide>).

- Expanded View files (replacing Supplementary Information)

Further information is available in our Guide For Authors:

The revision must be submitted online within 90 days; please click on the link below to submit the revision online before 2nd Feb 2020.

Link Not Available

Referee #1:

Vazquez-Armedariz et al report a bronchioalveolar lung organoid model and demonstrate its potential utility for studying lung development, intercellular communication and viral infection. The authors report a number of interesting observations. However, in general, the characterization tends to be superficial and warrants further study. Similarly, the authors dismiss prior lung organoid models as recapitulating "limited morphological and cellular features of the lung." This model does contain proximal and distal lineages (as do a number of other recently described models), but the morphological similarity stops there. BALO lack many of the proximal and distal features of native lung (i.e., submucosal glands, cartilage rings, large diameter vessels, capillaries, airway smooth muscle). In this respect, the similarity to native lung is, at present, overstated and not supported by the data. Finally, the relevance of the model to human lung biology remains unknown. This is, in part, because BASCs, the epithelial starting material for BALO, have not been identified in human lungs. Lung development, intercellular communication and viral infection can all be studied in mice in vivo. What is severely lacking is an improved in vitro model of adult human lung.

1) The epithelial organization of the spheres is difficult to assess. Are AECI and AECII found in relatively normal proportions (there seem to be few AECII in Fig 2F)? Do these cells make appropriate epithelial junctions with each other? The structure in Fig 2G is confusing. Is this a single airway surrounded by mesenchyme? The authors say in the legend that dotted lines indicate alveoli. Is this a typo? Fig 1 H suggests the presence of basal cells in BALO. Is this epithelium pseudostratified? Indeed, the text line 207 indicated basal-like cells in the ultrastructure, but this is not shown in the Figures. The importance of basal cells in human airway epithelium (and relative scarcity in distal mouse airways) makes this a particularly relevant point. What are the proportions of ciliated, secretory and basal cells? Is this uniform in all "proximal" BALO regions across multiple

organoids?

2) The characterization of mesenchymal lineages is exceptionally superficial. The cells are isolated based on the expression of *Pdgfra* and *Sca1* - both of which define heterogeneous populations. Were any *Sca1*⁺ epithelial cells observed? There seem to be very few *Pdgfra* negative cells in the *Sca1* intermediate and *Sca1*^{hi} sorts. Was single cell RNA seq or any other characterization performed on these freshly isolated mesenchymal cells? The y axes in Fig 3C are dramatically different and make direct comparison difficult. The differential effects of fibroblast subsets are BALO are superficial. What was the effect of each fraction on epithelial colony forming efficiency? Size of BALO? And the differentiation of alveolar and airway lineages?

3) There are no statistics to support enrichment of MYO or LIF genes in BALO-associated mesenchymal subpopulations (Fig 3E). In fact, many of the genes look equally expressed across the populations (i.e., *Axin2*, *Apoe*). It has been shown that MYO-LIPO are interconvertible lineages. Is this true in BALO?

4) Are other recently described fibroblast markers (e.g., *Lgr5*, *Lgr6*, *Fgf10*) observed in BALO-associated mesenchymal cells?

5) The organoids in Fig 3A and 3B look different from the organoids in other figures. The aSMA⁺ "MYO" are not seen in the alveolar region as they are in vivo and the distribution of aSMA⁺ cells is also not consistent.

6) The authors claim that TR-Macs are the "most relevant" but this is not supported by data or references. Circulating/recruited/interstitial myeloid lineages are equally likely to influence lung biology. Moreover, sessile macrophages, which make gap junction mediated contact with alveolar epithelial cells are not recovered in BALF. Are the macrophages observed after culture in BALO still alive? The ultrastructure in Fig 4D is not clear. Is the AEC an AECI or AECII? What is the evidence the TR-Mac is living and that the cells is a macrophage? Is there evidence that a lavagable TR-Mac should make contacts with either AECI or AECII? What is the nature of this contact? Is it passive or active? Could this be a macrophage embedded within lamellar body surfactant containing debris?

7) The staining in Fig 4E is not convincing. The connexin 43 staining appears cytoplasmic in the TR_MAC and not localized to any region of contact with surrounding cells. This must be shown by ultrastructure.

8) In Fig 4F the rMC seem uniform for expression of *Col1a1*, *Apoe*, and nearly all genes. Why is there no evidence of MYO and LIF? Similar to above, there are many basal cells in these BALO. Were these observed within BALO and what is their distribution? The relevance/importance of the heatmap in Fig 4F is unclear. It seems like a better analysis might be differential gene expression between each BALO population with and without TR Macs. Finally, were TR-Macs recovered from BALO and, if so, why weren't they sequenced? The claim that total differentiated cell number is increased after culture with TR-Macs is misleading. Were all cell types increased? Or was this a general increase in cell number? The data in Fig 4H does not support this claim (e.g., there appear to be fewer basal cells after culture with TR-Macs). The negative control for these experiments seems to be the absence of any injected cells. A more appropriate control to discern the effects of the purported "most relevant" macrophage population would be an irrelevant macrophage population or at least other cell type. The word "engraft" in line 316 and "defined functions" in 317 are not appropriate and not supported by data.

9) It is not clear why the control D11 BALO in Fig 5B are not branched.

10) The authors claim that there is less Bgal activity (and therefore Wnt signaling) after mo142-3p in BALO. This could just be an artifact of having fewer cells in general. What is the effect of mo142-3p on colony forming efficiency? And differentiation of airway/alveolar within BALO? And organization? Indeed, the lysotracker+ cells in Fig 3E seem to be centrally localized in BALO after mo142-3p.

Referee #2:

In this manuscript, Vazquez-Armendariz et al., have used a complex three-dimensional (3D) murine organoid model called bronchioalveolar lung organoid (BALO) to study the properties of EpCAM^{high}CD24^{low}Sca-1⁺ cells in the lung. Specifically, the authors purified EpCAM^{high}CD24^{low}Sca-1⁺ cells and co-cultured them with resident mesenchymal cell (rMC) to study the potential of these cells. The authors claim that these sorted cells generate a highly branched 3D organoid structure within 21 days, which mimics the cellular composition of the bronchioalveolar compartment. More importantly, the authors used two newly generated mouse models that express fluorescent reporters that allowed them to isolate SCGB1A1⁺SFTPC⁺ bronchioalveolar stem cells (BASCs). Using this model, the authors claim that of all the organoids derived from EpCAM^{high}CD24^{low}Sca-1⁺ cells predominantly originated from BASCs. Furthermore, using single cell RNA (scRNA-seq) sequencing, the authors claim that BALO consists of basal, secretory, ciliated, AEC II, and AEC I cell types. Additionally, the authors have used electron microscopy (EM) to demonstrate that BALO's form airway-like ultrastructure consists of basal-like cells, differentiated secretory cells filled with secretory granules, ciliated cells with mature cilia and basal bodies aligned underneath the apical cell surface, and alveolar type-1 and type-2 cells. Finally, the authors claim that their BALO system can be used for epithelial-immune cell interactions as well as modeling viral responses after influenza virus infection.

Overall, the data presented in this manuscript would be of great interest to the lung community. The experiments described here are well designed but there are some critical questions that require additional analysis/clarification.

Major comments:

1. The authors used a combination of EpCAM^{high}CD24^{low}Sca-1⁺ cells and SCGB1A1⁺SFTPC⁺ cells to setup organoid cultures. One of the major claims in this manuscript is that majority of the organoids in their BALO cultures were derived from SCGB1A1⁺SFTPC⁺ population and that they can have multilineage potential, meaning that they can generate basal, ciliated, club, alveolar type 2 and type 1 cells. The authors used scRNA-seq analysis, electron microscopy and immunostaining for cell type specific markers to show different cell types. However, it is unclear whether the authors used EpCAM^{high}CD24^{low}Sca-1⁺ cells or SCGB1A1⁺SFTPC⁺ cells for scRNA-seq and EM analysis. It is important in the context of recent finding (Liu et al., 2019 Nature Genetics) that BASCs can generate both airway and alveolar cells. However, Liu et al., 2019 have not found basal cells when they lineage traced BASCs in vivo. Therefore, it would be important to test whether SCGB1A1⁺SFTPC⁺ cells can generate basal cells, as the current manuscript claims? If yes, does it occur only ex vivo or can this be observed in vivo? The authors could use their newly developed BASC lineage tracing model or even simply one could use Sftpc-creER mouse line to test this?

2. The authors claim that the majority of the BALO's are derived SCGB1A1+SFTPC+ population which is about 5% of all EpCAM^{high}CD24^{low}Sca-1+ cells. The authors performed quantification of organoid forming ability of SCGB1A1+SFTPC+ population compared to SPC+ cells or SCGB1A1+ cells alone. However, the authors sorted mCherry+/YFP+ cells from EpCAM^{high}CD24^{low}Sca-1+ pre-gated cells. It would be important to check the colony forming efficiency of SCGB1A1+SFTPC+ population compared to SPC+ cells or SCGB1A1+ cells alone from total lung dissociates (not from EpCAM^{high}CD24^{low}Sca-1+ pre-gated cells). This is important to address as previous studies have showed that both SCGB1A1+ as well as SFTPC+ cells alone have the ability to generate organoids.

Minor comments:

3. In Fig1C SCGB1A1 and SFTPC staining's are not convincing. The authors could do better than this.

4. In Fig2G β -IV tubulin staining is not convincing. This can be improved.

Referee #3:

This study appears to be the first demonstration of an organoid model for the bronchiole/alveolar region of the lung. Robustness of these structures seems well documented in Figures 1-4. Additionally compelling is the addition of distinct mesenchymal subpopulations (PDGFR α high versus low fibroblasts) which direct distal versus more proximal structural fates.

Evidence that macrophages can be injected into the lumen of these structures, with apparent pro-epithelial differentiation function is remarkable. The system is also amenable to infection by flu, and knock-down analyses for parsing molecular mechanism.

Overall, the figures/data and writing is very clear. The study makes a substantial advance over previous organoid models (which largely focus on airway or alveolospheres alone).

A minor quibble is that efforts to show all possible applications of the BALO model led to some of the analyses being superficial (e.g., effect of macrophages on epithelial differentiation inferred from scRNAseq; use of Wnt reporters nice but not clear how reproducible these findings are, as only single organoids are shown without quantification). So while I might have preferred greater depth of analysis for these figures-- I do concede that the main finding/technology is exciting and should be share with the community without delay.

Minor:

1. Fig.4D is not compelling, as the filopodial projections/contacts between Macs/AT2 cells are not clear. Also, can the authors comment on whether addition of macrophages had consequences on surfactant uptake in the lumen?

Point-by-point response to comments from**Referee #1:**

Vazquez-Armendariz et al report a bronchioalveolar lung organoid model and demonstrate its potential utility for studying lung development, intercellular communication and viral infection. The authors report a number of interesting observations. However, in general, the characterization tends to be superficial and warrants further study. Similarly, the authors dismiss prior lung organoid models as recapitulating "limited morphological and cellular features of the lung." This model does contain proximal and distal lineages (as do a number of other recently described models), but the morphological similarity stops there. BALO lack many of the proximal and distal features of native lung (i.e., submucosal glands, cartilage rings, large diameter vessels, capillaries, airway smooth muscle). In this respect, the similarity to native lung is, at present, overstated and not supported by the data. Finally, the relevance of the model to human lung biology remains unknown. This is, in part, because BASCs, the epithelial starting material for BALO, have not been identified in human lungs. Lung development, intercellular communication and viral infection can all be studied in mice *in vivo*. What is severely lacking is an improved *in vitro* model of adult human lung.

Response: The authors thank the reviewer for the opportunity to discuss these important points. It was not our intention to claim that bronchioalveolar lung organoids' (BALO) structure exactly resembles the adult mouse lung. However, we feel that this model, among the lung organoid models published to date (Barkauskas et al., 2013; Lee et al., 2014; Rock & Hogan, 2011), is most advanced concerning both 3D structure and cellular composition of the epithelial and mesenchymal compartments. Particularly, we meant to highlight the similarity in terms of cell type diversity and distribution of these cells along the proximo-distal axis. For example, matrix fibroblasts/lipofibroblasts (LIF) were closely associated with type II alveolar epithelial cells (AEC II) while myofibroblasts (MYO) provided the scaffold for BALO patterning, as known for the lung *in vivo*. To our knowledge, this is the first organoid system to reflect such cellular heterogeneity and spatial distribution of the various bronchiolar and alveolar epithelial cells (basal cells, club/secretory cells, ciliated cells, AEC I, AEC II) as well as mesenchymal cells (LIF and MYO). Further, electron microscopy (EM) analyses detailed regions in BALO resembling bronchoalveolar duct junctions (bronchiolar and intermediate cells types) that lead into alveolar regions with flattened AEC I (**Appendix Figure S2B**), not reported in other lung organoid models to date. We would like to point out that no other current lung organoid model derived from murine adult stem cells has clearly proven the presence of both AEC I and mature AEC II organized in an alveolar-like configuration associated to tubular airway structures, making our model much more suitable to study alveolar formation and disease than other *in vitro* models. Until now, bronchioalveolar stem cells (BASC) have not yet been identified in the human lungs, nevertheless, the findings uncovered by our collaborators in Salwig *et al.*, 2019 along with further genomic profiling and lineage tracing of the BASC found in BALO may reveal unique markers for BASC identification *in vivo* in mice that could allow the discovery of a corresponding BASC population in the human lung (Salwig *et al.*, 2019). We have included this issue in the discussion section on **page 20 lines 428-432** of the revised manuscript. Nevertheless, the BALO model allows introduction of immune cells into BALO by microinjection where single defined cell populations (as shown for TR-Mac) of wildtype (WT) or gain-/loss-of-function phenotype can be added to the system. This is also a

suitable tool to study the role of defined cell populations and genes/molecules expressed in these cells, which is often difficult to achieve *in vivo* due to the lack of tools that would allow targeting a very specific subset of cells *in vivo*. Such cells could be flow-sorted from WT or transgenic mice and introduced into the BALO. Therefore, we believe that in the context of previously published *in vitro* models, our BALO system represents an important forward step towards establishing more comprehensive *in vitro* models for studying developmental and disease processes, particularly mechanisms involving epithelial-mesenchymal-myeloid unit interactions that are at the moment difficult to study *in vivo*.

- 1) The epithelial organization of the spheres is difficult to assess. Are AEC I and AEC II found in relatively normal proportions (there seem to be few AEC II in Fig 2F)? Do these cells make appropriate epithelial junctions with each other? The structure in Fig 2G is confusing. Is this a single airway surrounded by mesenchyme? The authors say in the legend that dotted lines indicate alveoli. Is this a typo? Fig 1 H suggests the presence of basal cells in BALO. Is this epithelium pseudostratified? Indeed, the text line 207 indicated basal-like cells in the ultrastructure, but this is not shown in the Figures. The importance of basal cells in human airway epithelium (and relative scarcity in distal mouse airways) makes this a particularly relevant point. What are the proportions of ciliated, secretory and basal cells? Is this uniform in all "proximal" BALO regions across multiple organoids?

Response: The authors thank the reviewer for raising these points. We apologize for the confusing figure; indeed, AEC I and AEC II are found in normal ("*in vivo*") proportions within BALO. We have made changes in **Figure 2F (now new Figure EV2A)** to better illustrate BALO's alveolar-like structures comprised of cuboidal AEC II (SFTPC⁺ cells) and flat podoplanin (PDPN)⁺ AEC I. Further, the proportions of AEC II (94.3±0.2%) and the low-frequency PDPN⁺ AEC I (5.3±0.6%) subpopulations are shown by FACS analyses in **Appendix Figure S2A** and are now mentioned in the results section on **page 9 lines 189-191**. Indeed, proper tight junctions between AECs within BALO's alveolar-like structures are observed and are now depicted by EM in **new Figure 2B and new Appendix Figure S2E** of the revised manuscript. Regarding **Figure 2G (now new Figure EV2B)**, the mentioned typo has been corrected in the figure legend of the revised manuscript. Furthermore, we have added a new confocal picture showing the presence of ciliated and club cells within BALO's airway-like structures in **new Figure EV2B** which can also be observed in **new Figure 2I**. Indeed, as stated by the reviewer, airway-like structures in day 40 BALO form a pseudostratified-like epithelium which is also depicted in **new Figure 2I** by EM. To make this clear, we have expanded the figure legend and added more labels which include basal-like cells within BALO's airway-like structures. Further, our single cell RNA-sequencing (ScRNA-seq) data revealed that BALO's airway cell proportions of the club/secretory cells (79±9.5%), ciliated cells (2.8±1.9%), and basal cells (18.2±8.8%) remain constant across different cultures analyzed at day 23 (n=6). Consequently, while we acknowledge that our system does not possess all the proximal and distal characteristics of native lung such as a vascular component, we think that this model represents a step forward towards creating more comprehensive and complex systems for modeling lung biology and disease *in vitro*.

- 2) The characterization of mesenchymal lineages is exceptionally superficial. The cells are isolated based on the expression of *Pdgfra* and *Sca1* - both of which define heterogeneous populations. Were any *Sca1*⁺ epithelial cells observed? There seem to be very few *Pdgfra* negative cells in the *Sca1* intermediate and *Sca1*^{hi} sorts. Was single cell RNA seq or any other characterization performed on these freshly isolated mesenchymal cells? The y axes in Fig 3C are dramatically different and make direct comparison difficult. The differential effects of fibroblast subsets are BALO are superficial. What was the effect of each fraction on epithelial colony forming efficiency? Size of BALO? And the differentiation of alveolar and airway lineages?

Response: Indeed, as others have shown (McQualter et al., 2009), resident mesenchymal cells (rMC) can be isolated based on the cell surface expression of platelet-derived growth factor receptor alpha (PDGFR α), stem cells antigen-1 (*Sca-1*), and lack of expression of epithelial cell markers (eg EpCAM), which in fact represents a heterogeneous cell population. Notably, as depicted in **Figure 1A**, expression of the surface marker *Sca-1* was also shown to be essential for the enrichment of BASC. Regarding further rMC characterization, this population heterogeneity was confirmed by Sc-RNA seq analysis of freshly isolated rMC (defined as *Sca-1*⁺PDGFR α ⁺), which revealed six distinct cell populations identified as myofibroblast (MYO)-like, lipofibroblast (LIF)-like and four different subsets of stromal fibroblasts (**see reply Figure 1**). Accordingly, MYO and LIF rMC subsets were as well identified by Sc-RNA seq analysis of digested BALO in **Figure 3E** which were both shown to be necessary for proper BALO formation (**Figure 3D**). To assess the importance of each mesenchymal fraction, we used the *Pdgfra*^{GFP} knock-in mouse-line. As for **Figure 3C**, we agree with the reviewer's rationale that the y axes in the histograms are different but they were not meant to be compared in terms of cell numbers but rather to show the purity of the isolated cells populations. We would like to point out that, in the left panel of **Figure 3C**, *Sca-1*⁺ rMC are shown based on their PDGFR α expression where the proportion of each rMC subset can be compared. As shown in **Figure 3D**, co-culture of *Sca-1*^{int} PDGFR α ^{hi} rMC with BASC did not support the growth of organoids at all. In contrast, the *Sca-1*^{hi}PDGFR α ^{low} rMC fraction supported organoid formation at a colony forming unit (CFU) frequency of ~1:100 (**new Figure S3D**) but did not promote differentiation into BALO, whereas addition of both fractions revealed the same CFU frequency (1:100) but promoted BALO formation including alveolar differentiation and branching morphogenesis. This feature is now shown by confocal microscopy of SFTPC⁺ AEC II in **new Figure S3E** and mentioned in the results section on **page 12 lines 258-261** of the revised manuscript.

Figure for referees removed

- 3) There are no statistics to support enrichment of MYO of LIF genes in BALO-associated mesenchymal subpopulations (Fig 3E). In fact, many of the genes look equally expressed across the populations (i.e., *Axin2*, *Apoe*). It has been shown that MYO-LIPO are interconvertible lineages. Is this true in BALO?

Response: We thank the reviewer for pointing this out. We have addressed this point by including a table depicting the adjusted p-values of the mesenchymal genes associated with MYO (C3) or LIF (C4) phenotypes (**Table 1**). Of note, *Plin2* was not significantly upregulated in the LIF cluster (it is also expressed in epithelial cells); therefore, it was replaced for another well-known LIF marker, *Fgf10*, in **new Figure 3E** and **Table 1** of the revised manuscript (please see also response to question 4). Concerning the question whether MYO-LIF were interconvertible lineages, we would like to stress that the term myofibroblast (“MYO”) is widely used in the field to refer to mesenchymal cells that express the smooth muscle cell marker alpha smooth muscle actin (*ACTA2*). In the case of pulmonary fibrosis, MYO accumulate in the lung and produce collagen, thereby mediating fibrogenesis. We have shown that these cells derive from LIF and that when fibrosis resolves, MYO can revert back to LIF. Thus, this interconversion between LIF and MYO, as described in (El Agha et al., 2017; Kheirollahi et al., 2019), is a process happening in a particular pathological *in vivo* condition (pulmonary fibrosis development and resolution). To date, it is still unknown how heterogeneous the fibrosis-associated MYO population is and whether this population resembles other cell populations that are involved in non-pathological conditions, such as alveolar morphogenesis (e.g. alveolar MYO that transiently appear in the developing lung during alveologogenesis). In the current work, we did not intend to imply that the MYO population that we identified using ScRNA-seq resembles fibrosis-associated MYO (that derive from LIF). We apologize for the confusion. On the other hand, there is still ambiguity in the field regarding the nomenclature of the different subsets of mesenchymal cells in the lung, particularly for MYO. For instance, according to LungGENS (Du, Guo, Whitsett, & Xu, 2015) (single-cell RNA-seq database; part of the LungMAP project), MYO are listed as MYO/Smooth muscle (as a single entity) until postnatal day 14 and there is no mention of MYO beyond this stage. Therefore, further characterization of the MYO identity is still pending, but we feel that this is beyond the scope of this manuscript. Still, we appreciate that the reviewer brought this issue up. In order to avoid confusion of the readers, we have addressed this aspect to the discussion section on **pages 23 lines 495-498** of the revised manuscript.

- 4) Are other recently described fibroblast markers (e.g., *Lgr5*, *Lgr6*, *Fgf10*) observed in BALO-associated mesenchymal cells?

Response: We thank the reviewer for raising this important question. Notably, gene expression of *Fgf10* was detected in the LIF rMC subpopulation and has been included in **new Figure 3E** and **Table 1** of the revised manuscript. In our Sc-RNA seq analysis the fibroblast markers *Lrg5* and *Lrg6* were not detected in digested BALO after 21 days of

culture, nonetheless, we cannot exclude the possibility that these markers may be expressed in rMC at earlier time points during BALO formation. We have elaborated on this in the discussion section on **page 23 lines 501-503** of the revised manuscript

- 5) The organoids in Fig 3A and 3B look different from the organoids in other figures. The aSMA+ "MYO" are not seen in the alveolar region as they are *in vivo* and the distribution of aSMA+ cells is also not consistent.

Response: The authors apologize for the unclear description of the **Figures 3A and 3B**. For these figures we primarily aimed to focus on the LIF population surrounding the BALO's alveolar-like structures. However, the reviewer's point is clearly relevant for better understanding of the MYO spatial distribution; therefore, we have now included representative confocal pictures showing aSMA+ cells found in close association with cells present within the alveolar-like regions of BALO in **new Appendix Figure S3B** of the revised manuscript.

- 6) The authors claim that TR-Macs are the "most relevant" but this is not supported by data or references. Circulating/recruited/interstitial myeloid lineages are equally likely to influence lung biology. Moreover, sessile macrophages, which make gap junction mediated contact with alveolar epithelial cells are not recovered in BALF. Are the macrophages observed after culture in BALO still alive? The ultrastructure in Fig 4D is not clear. Is the AEC an AECI or AECII? What is the evidence the TR-Mac is living and that the cells is a macrophage? Is there evidence that a lavagable TR-Mac should make contacts with either AECI or AECII? What is the nature of this contact? Is it passive or active? Could this be a macrophage embedded within lamellar body surfactant containing debris?

Response: We thank the reviewer for these comments. We meant to highlight that introducing TR-Mac, which are considered the major sessile innate immune cell population and the first line of defense against incoming pathogens (Wynn & Vannella, 2016), would provide a more advanced system to study disease *in vitro*. Nonetheless, to be more precise in our wording, we have re-phrased the sentence and removed the "most relevant" term from the revised manuscript (**page 14 lines 304-308**). Notably, Westphalen *et al* showed that Connexin43 (Cx43) expression was detected in TR-Mac isolated from both lung tissue and bronchioalveolar lavage (BAL), although with a significant higher level in TR-Mac recovered directly from the lung homogenate (Westphalen et al., 2014). Therefore, given our data presented in **Figure 4**, we can infer that TR-Mac recovered from BAL still hold the capacity to increase Cx43 levels when introduced into BALO and make contact with AEC. In **Appendix Figure S4C**, most of the microinjected TR-Mac (>80%) remained after 10 days post-microinjection without losing their integrity. Nonetheless, to confirm TR-Mac viability, microinjected TR-Mac were stained with a viability dye showing that 86.7±7% of the TR-Mac were still viable 10 days post-microinjection into BALO. This is now shown in **new Appendix Figure S4D** and mentioned in the results section on **page 14 lines 313-315** of the revised manuscript. Furthermore, all the electron microscopy data and terminology presented in our manuscript were cautiously evaluated by Prof. Wolfgang Kummer, a lung anatomy expert with decades of experience in electron microscopy. In **Figure 4D**, the TR-Mac has an intact cellular membrane without signs of cytoplasm condensation which indicates its viability and is in contact with a thin and elongated AEC I. TR-Mac were identified based on the presence of an

actin filament bundle characteristic of TR-Mac (Trotter, 1981) which is now depicted in **new Figure 4D** of the revised manuscript. In addition, we have now included new representative EM pictures in **new Figure 4D** of the revised manuscript to further illustrate macrophage interactions with AEC.

- 7) The staining in Fig 4E is not convincing. The connexin 43 staining appears cytoplasmic in the TR-Mac and not localized to any region of contact with surrounding cells. This must be shown by ultrastructure.

Response: The authors apologize for the unclear staining. Until now, ultrastructural demonstration of gap junctions such as Cx43 was found to be difficult and has been shown only at some particular examples such as between smooth muscle cells and at the apical junctional complex in certain epithelia (Al-Ghadban, Kaissi, Homaidan, Naim, & El-Sabban, 2016). A study showing gap junctions between macrophages and lung epithelial cells has been performed in flat 2D cultures that require freeze-fracture electron microscopy and immunogold-labelling which cannot be well applied to 3D structures such as BALO. Notably, other studies have shown that Cx43 is also observed within the cell's cytoplasm during its biosynthesis (Beckmann, Grissmer, Meier, & Tschernig, 2020). However, we agree with the reviewer's comment that a clear region of contact between TR-Mac and AEC was difficult to assess in the image we presented in Figure 4E. Therefore, to address the reviewer's point, the image in **new Figure 4E** has been now replaced to better illustrate expression of Cx43 in TR-Mac and AEC in close association within BALO.

- 8) In Fig 4F the rMC seem uniform for expression of Col1a1, Apoe, and nearly all genes. Why is there no evidence of MYO and LIF? Similar to above, there are many basal cells in these BALO. Were these observed within BALO and what is their distribution? The relevance/importance of the heatmap in Fig 4F is unclear. It seems like a better analysis might be differential gene expression between each BALO population with and without TR Macs. Finally, were TR-Macs recovered from BALO and, if so, why weren't they sequenced? The claim that total differentiated cell number is increased after culture with TR-Macs is misleading. Were all cell types increased? Or was this a general increase in cell number? The data in Fig 4H does not support this claim (e.g., there appear to be fewer basal cells after culture with TR-Macs). The negative control for these experiments seems to be the absence of any injected cells. A more appropriate control to discern the effects of the purported "most relevant" macrophage population would be an irrelevant macrophage population or at least other cell type. The word "engraft" in line 316 and "defined functions" in 317 are not appropriate and not supported by data.

Response: For the Sc-RNA seq experiment shown in **Figure 1F**, sample preparation focused on the identification of epithelial versus mesenchymal cell populations present in BALO cultures. Therefore, we used a strategy where only a few hundred cells were subjected to single cell RNA sequencing. However, for the Sc-RNA seq experiment performed in **Figure 4F**, higher cell numbers were processed in a way to achieve a high yield of epithelial cells, allowing a deeper analysis to reveal epithelial subpopulations and confirming the presence of basal cells (*Krt5*, *Krt14*, *Aqp3*, *Trp63*), club/secretory cells (*Scgb3a2*, *Muc5b*, *Bpifa1*), ciliated cells (*Foxj1*, *Tppp3*, *Lrrc23*), AECI (*Hopx*, *Ager*, *Cldn18*) and AECII (*Sftpc*, *Lyz2*, *Sftpb*), but resulting in low numbers of rMC and therefore,

no resolution of rMC subsets. As mentioned in the response for question #1, the basal cells are located at the basal side of BALO's airway-like structures forming a pseudostratified epithelium (**new Figure 2I**). In **Figure 4F**, we primarily aimed to evaluate the effects of TR-Mac on the BALO epithelium while the comparative differential gene expression between each BALO population with and without TR-Mac is depicted in **Figure 4G** where genes involved in cell proliferation and differentiation showed significant differences when TR-Mac had been introduced. As mentioned, TR-Mac *per se* were not the focus of the experiment and the digestion protocol was set up to favor epithelial cell isolation resulting in a low number of TR-Mac to be sequenced. In **Figure 4H**, we meant to show an increase in epithelial cell differentiation when TR-Mac were present within BALO, nonetheless, we agree with the reviewer's point and to be clearer, we have now compared only terminally differentiated cells (AECI and ciliated cells) which is now shown in **new Figure 4H**. We respectfully disagree with the reviewer's comment regarding the use of the terms "engraft" and "defined functions". Our data have revealed that microinjected TR-Mac are alive, remain within BALO for at least 10 days and directly interact with AEC as shown by Cx43 staining and EM morphology which supports our claim on TR-Mac engraftment into BALO's alveoli. Moreover, our Sc-RNA seq data indicates transcriptional changes in genes involved in cell proliferation and differentiation on BALO epithelium when TR-Mac are present. In addition, TR-Mac raise an innate immune response against viral infection as shown in **Figure 6G**, and take up and digest surfactant (please see below the response to Reviewer 3) indicating that TR-Mac provide defined functions in BALO.

9) It is not clear why the control D11 BALO in Fig 5B are not branched?

Response: We apologize for the unclear description of the method. Indeed, under normal culture conditions, the differentiation into BALO starts after 10 to 11 days of culture. However, for proper transfection efficiency during loss-of-function experiments, both scrambled control and mo142-3p were diluted in media containing a concentration of 0.2% FCS instead of 10% which led to a slight delay in BALO differentiation. We have rephrased this sentence in the revised manuscript to make this particular point clear for the reader in the results section on **page 16 lines 354-358**.

10) The authors claim that there is less Bgal activity (and therefore Wnt signaling) after mo142-3p in BALO. This could just be an artifact of having fewer cells in general. What is the effect of mo142-3p on colony forming efficiency? And differentiation of airway/alveolar within BALO? And organization? Indeed, the lysotracker+ cells in Fig 3E seem to be centrally localized in BALO after mo142-3p.

Response: We thank the reviewer for this important remark. **Appendix Figure S5C** shows that adenomatous polyposis coli (*Apc*), a key negative regulator of the WNT signaling pathway, is upregulated in BALO cultures after treatment with mo142-3p compared to scramble control. This effect was shown by qPCR which determines the average expression of *Apc* per cell, thus, the chance of bias due to a differential number of cells in mo142-3p-treated BALO is minimal. Nevertheless, we have further addressed this particular issue by including the colony forming efficiency of control (scramble)- and morpholino-treated cultures in **new Appendix Figure S5A** and mention it in the results section on **page 16 line 358-361** of the revised manuscript. There was no direct effect on

the CFU after addition of mo142-3p when the treatment with morpholino or scramble control was started on day 6 of culture. As suggested by the reviewer, evaluation of airway and alveolar differentiation in BALO was done fifteen days after treatment with mo142-3p or scramble by FACS and has now been included in **new Appendix Figure S5D** and stated in the results section on **page 17 lines 378-379**. A significant decrease of AEC II (EpCAM⁺CD49^{int}Lyzotracker⁺) and club/ secretory cells (EpCAM⁺CD49^{high}CD24^{low}) percentage in mo142-3p treated cultures compared to scramble controls was observed confirming the importance of the WNT signaling on BALO differentiation rather than early colony outgrowth.

Referee #2:

In this manuscript, Vazquez-Armendariz et al., have used a complex three-dimensional (3D) murine organoid model called bronchioalveolar lung organoid (BALO) to study the properties of EpCAM^{high}CD24^{low}Sca-1⁺ cells in the lung. Specifically, the authors purified EpCAM^{high}CD24^{low}Sca-1⁺ cells and co-cultured them with resident mesenchymal cell (rMC) to study the potential of these cells. The authors claim that these sorted cells generate a highly branched 3D organoid structure within 21 days, which mimics the cellular composition of the bronchioalveolar compartment. More importantly, the authors used two newly generated mouse models that express fluorescent reporters that allowed them to isolate SCGB1A1⁺SFTPC⁺ bronchioalveolar stem cells (BASCs). Using this model, the authors claim that of all the organoids derived from EpCAM^{high}CD24^{low}Sca-1⁺ cells predominantly originated from BASCs. Furthermore, using single cell RNA (scRNA-seq) sequencing, the authors claim that BALO consists of basal, secretory, ciliated, AEC II, and AEC I cell types. Additionally, the authors have used electron microscopy (EM) to demonstrate that BALO's form airway-like ultrastructure consists of basal-like cells, differentiated secretory cells filled with secretory granules, ciliated cells with mature cilia and basal bodies aligned underneath the apical cell surface, and alveolar type-1 and type-2 cells. Finally, the authors claim that their BALO system can be used for epithelial-immune cell interactions as well as modeling viral responses after influenza virus infection. Overall, the data presented in this manuscript would be of great interest to the lung community. The experiments described here are well designed but there are some critical questions that require additional analysis/clarification.

Response: The authors thank the reviewer for the positive comments and the opportunity to address the mentioned concerns.

Major comments:

- 1) The authors used a combination of EpCAM^{high}CD24^{low}Sca-1⁺ cells and SCGB1A1⁺SFTPC⁺ cells to setup organoid cultures. One of the major claims in this manuscript is that majority of the organoids in their BALO cultures were derived from SCGB1A1⁺SFTPC⁺ population and that they can have multilineage potential, meaning that they can generate basal, ciliated, club, alveolar type 2 and type 1 cells. The authors used scRNA-sq analysis, electron microscopy and immunostaining for cell type specific markers to show different cell types. However, it is unclear whether the authors used EpCAM^{high}CD24^{low}Sca-1⁺ cells or SCGB1A1⁺SFTPC⁺ cells for ScRNA-seq and EM analysis. It is important in the context

of recent finding (Liu et al., 2019 Nature Genetics) that BASCs can generate both airway and alveolar cells. However, Liu et al., 2019 have not found basal cells when they lineage traced BASCs *in vivo*. Therefore, it would be important to test whether SCGB1A1⁺SFTPC⁺ cells can generate basal cells, as the current manuscript claims? If yes, does it occur only *ex vivo* or can this be observed *in vivo*? The authors could use their newly developed BASC lineage tracing model or even simply one could use Sftpc-creER mouse line to test this?

Response: The authors thank the reviewer for this important comment. The ScRNA-seq and EM analysis were performed using the EpCAM^{high}CD24^{low}Sca-1⁺ population enriched in BASC. Indeed, Lui *et al* did not find basal cells when BASC were traced *in vivo*. Therefore, to further confirm that our double positive BASC are capable to give rise to basal cells in BALO, we isolated SCGB1A1⁺SFTPC⁺ BASC from the lungs of BASC v-race (SPC-2A-YFP-2A-tTA-N, CCSP-2A-mCherry-2A-tTA-C, tetO^{biluc}/Cre, Rosa26^{stopfllox-lacZ}) mice (Salwig et al., 2019) in which reconstituted split-tTA effectors induce expression of Cre recombinase and permanently label BASC and all their descendants via β -galactosidase activity. The **new Appendix Figure S1H** shows that after 21 days of culture, all cells within a BALO were LacZ⁺ indicating that all BALO cells including basal cells are descendants from BASC. Importantly, even upon higher magnification, LacZ^{neg} cells could not be observed. These data are now mentioned in the results section on **page 7 lines 152-154** of the revised manuscript. As advised by the reviewer, we also tested SCGB1A1⁺SFTPC⁺ BASC differentiation into basal cells *in vivo* after influenza-induced lung injury. BASC v-race mice were on doxycycline treatment to exclude de novo labeling of BASC and lungs were isolated and labeled 35 days post-infection. After β -galactosidase labeling, basal cell marker krt5 co-expression with labeled cells (i.e. BASC that were present before infection plus all their descendants emerging during regeneration) was evaluated by immunohistochemistry (IHC) (**Reply Figure 2**). Clusters of β -gal⁺ cells were observed in the terminal bronchioles and alveoli, however, co-localization with krt5⁺ basal cells or basal cell-like pods were not detected, indicating that BASC do not differentiate into basal cells *in vivo*. Therefore, we hypothesize that BASC differentiation into basal cells observed in BALO seems to be an *in vitro* artifact that might be, for example, driven by the total absence of basal cells in the BALO airway-like niche which is not the case *in vivo*. We have now addressed this aspect in the discussion section on **page 21-22 lines 468-471** of the revised manuscript.

Figures for referees removed

- 2) The authors claim that the majority of the BALO's are derived SCGB1A1⁺SFTPC⁺ population which is about 5% of all EpCAM^{high}CD24^{low}Sca-1⁺ cells. The authors performed quantification of organoid forming ability of SCGB1A1⁺SFTPC⁺ population compared to SPC⁺ cells or SCGB1A1⁺ cells alone. However, the authors sorted mCherry⁺/YFP⁺ cells from EpCAM^{high}CD24^{low}Sca-1⁺ pre-gated cells. It would be important to check the colony forming efficiency of SCGB1A1⁺SFTPC⁺ population compared to SPC⁺ cells or SCGB1A1⁺ cells alone from total lung dissociates (not from EpCAM^{high}CD24^{low}Sca-1⁺ pre-gated cells). This is important to address as previous studies have showed that both SCGB1A1⁺ as well as SFTPC⁺ cells alone have the ability to generate organoids.

Response: The authors agree with the reviewer's rationale. We are in fact excluding most of the SFTPC⁺ AEC II with our gating strategy (with the intention to prevent contamination from these cells to increase purity of the sorted double+ cells), as well as a fraction of the single⁺ SCGB1A1⁺ club cells. As suggested by the reviewer, we approached this issue by modifying our gating strategy to sort all SFTPC⁺ and SCGB1A1⁺ cells directly from the whole EpCAM⁺ population without the addition of further surface markers. Using this protocol, we revealed that compared to the 1:100 CFU of the SCGB1A1⁺SFTPC⁺ population, the colony forming efficiency of SFTPC⁺ AECII and SCGB1A1⁺ club cells were overall 1:25 and 1:1000, respectively. Notably, SFTPC⁺ and SCGB1A1⁺ cells isolated from the whole EpCAM⁺ fraction formed only alveolospheres (Barkauskas et al., 2013) and bronchiolospheres (Lee et al., 2014), respectively. BALO with proximo-distal cell specification were only observed in cultures derived from SCGB1A1⁺SFTPC⁺ cells. These results were included in the revised manuscript in **new Appendix Figure S1G** and we refer to the different colony-forming efficiencies in the results section on **page 7 line 137-145** of the revised manuscript.

Minor comments:

- 3) In Fig1C SCGB1A1 and SFTPC staining's are not convincing. The authors could do better than this.

Response: We agree with the reviewer's point. We have moved **Figure 1C** to the supplement (**new Appendix Figure 1A**) and included in the revised manuscript a new confocal picture and a 3D reconstruction of BALO derived from EpCAM^{high}CD24^{low}Sca-1⁺SCGB1A1⁺SFTPC⁺ cells isolated from Scgb1a1^{mCherrySftpc}^{YFP} reporter mice which much better depicts SCGB1A1⁺ and SFTPC⁺ cells localization and distribution (**new Figure EV1A and EV1B**).

4) In Fig2G β -IV tubulin staining is not convincing. This can be improved.

Response: The authors thank the reviewer for this comment. We have changed **Figure 2G (now new Figure EV2B)** in the revised manuscript to better illustrate the presence of multiple ciliated cells interspersed by club cells within the BALO airway-like structures.

Referee #3:

This study appears to be the first demonstration of an organoid model for the bronchiole/alveolar region of the lung. Robustness of these structures seems well documented in Figures 1-4. Additionally, compelling is the addition of distinct mesenchymal subpopulations (PDGFRa high versus low fibroblasts) which direct distal versus more proximal structural fates. Evidence that macrophages can be injected into the lumen of these structures, with apparent pro-epithelial differentiation function is remarkable. The system is also amenable to infection by flu, and knock-down analyses for parsing molecular mechanism. Overall, the figures/data and writing is very clear. The study makes a substantial advance over previous organoid models (which largely focus on airway or alveolospheres alone). A minor quibble is that efforts to show all possible applications of the BALO model led to some of the analyses being superficial (e.g., effect of macrophages on epithelial differentiation inferred from scRNAseq; use of Wnt reporters nice but not clear how reproducible these findings are, as only single organoids are shown without quantification). So while I might have preferred greater depth of analysis for these figures-- I do concede that the main finding/technology is exciting and should be shared with the community without delay.

Response: The authors thank the reviewer for the positive feedback on our model. We have now included additional quantification and cell differentiation data on scramble and morpholino-treated organoids (please see Reviewer 1 response 10).

Minor:

1. Fig.4D is not compelling, as the filopodial projections/contacts between Macs/AT2 cells are not clear. Also, can the authors comment on whether addition of macrophages had consequences on surfactant uptake in the lumen?

Response: The authors agree with the reviewer's point. Therefore, we have included a new EM picture showing contact between a macrophage and AEC within BALO's alveolar-like structures in **new Figure 4D** of the revised manuscript. Prof. Wolfgang Kummer, a lung anatomy expert, analyzed the EM data and confirmed that TR-Mac-AEC contacts are indeed observed in BALO, and that the provided EM pictures are representative for what can be observed in BALO reconstituted by TR-Mac. Indeed, uptake of surfactant and its digestion (i.e. whirls of lamellar surfactant embraced together within the cell and not separated in individual lamellar bodies) by TR-Mac was observed by EM (as verified by W. Kummer). This is now shown in **new Figure 4D** and **new Appendix Figure S4E** and mentioned in the results section on **page 15 line 319-321** of the revised manuscript.

References

- Al-Ghadban, S., Kaissi, S., Homaidan, F. R., Naim, H. Y., & El-Sabban, M. E. (2016). Cross-talk between intestinal epithelial cells and immune cells in inflammatory bowel disease. *Scientific Reports*, *6*, 29783. <https://doi.org/10.1038/srep29783>
- Barkauskas, C. E., Crouce, M. J., Rackley, C. R., Bowie, E. J., Keene, D. R., Stripp, B. R., ... Hogan, B. L. M. (2013). Type 2 alveolar cells are stem cells in adult lung. *The Journal of Clinical Investigation*, *123*(7), 3025–3036. <https://doi.org/10.1172/JCI68782>
- Beckmann, A., Grissmer, A., Meier, C., & Tschernig, T. (2020). Intercellular communication between alveolar epithelial cells and macrophages. *Annals of Anatomy*, *227*, 151417. <https://doi.org/10.1016/j.aanat.2019.151417>
- Du, Y., Guo, M., Whitsett, J. A., & Xu, Y. (2015). “LungGENS”: a web-based tool for mapping single-cell gene expression in the developing lung. *Thorax*, *70*(11), 1092–1094. <https://doi.org/10.1136/thoraxjnl-2015-207035>
- El Agha, E., Moiseenko, A., Kheirollahi, V., De Langhe, S., Crnkovic, S., Kwapiszewska, G., ... Bellusci, S. (2017). Two-Way Conversion between Lipogenic and Myogenic Fibroblastic Phenotypes Marks the Progression and Resolution of Lung Fibrosis. *Cell Stem Cell*, *20*(2), 261-273.e3. <https://doi.org/10.1016/j.stem.2016.10.004>
- Kheirollahi, V., Wasnick, R. M., Biasin, V., Vazquez-Armendariz, A. I., Chu, X., Moiseenko, A., ... El Agha, E. (2019). Metformin induces lipogenic differentiation in myofibroblasts to reverse lung fibrosis. *Nature Communications*, *10*(1), 1–16. <https://doi.org/10.1038/s41467-019-10839-0>
- Lee, J. H., Bhang, D. H., Beede, A., Huang, T. L., Stripp, B. R., Bloch, K. D., ... Kim, C. F. (2014). Lung stem cell differentiation in mice directed by endothelial cells via a BMP4-NFATc1-thrombospondin-1 axis. *Cell*, *156*(3), 440–455. <https://doi.org/10.1016/j.cell.2013.12.039>
- McQualter, J. L., Brouard, N., Williams, B., Baird, B. N., Sims-Lucas, S., Yuen, K., ... Bertoncello, I. (2009). Endogenous Fibroblastic Progenitor Cells in the Adult Mouse Lung Are Highly Enriched in the Sca-1 Positive Cell Fraction. *Stem Cells*, *27*(3), 623–633. <https://doi.org/10.1634/stemcells.2008-0866>
- Rock, J. R., & Hogan, B. L. M. (2011). Epithelial progenitor cells in lung development, maintenance, repair, and disease. *Annual Review of Cell and Developmental Biology*, *27*, 493–512. <https://doi.org/10.1146/annurev-cellbio-100109-104040>
- Salwig, I., Spitznagel, B., Vazquez-Armendariz, A. I., Khalooghi, K., Guenther, S., Herold, S., ... Braun, T. (2019). Bronchioalveolar stem cells are a main source for regeneration of distal lung epithelia *in vivo*. *The EMBO Journal*, *38*(12), e102099. <https://doi.org/10.15252/emj.2019102099>
- Trotter, J. A. (1981). The organization of actin in spreading macrophages. The actin-cytoskeleton of peritoneal macrophages is linked to the substratum via transmembrane connections. *Experimental Cell Research*, *132*(2), 235–248. [https://doi.org/10.1016/0014-4827\(81\)90099-9](https://doi.org/10.1016/0014-4827(81)90099-9)
- Westphalen, K., Gusarova, G. A., Islam, M. N., Subramanian, M., Cohen, T. S., Prince,

A. S., & Bhattacharya, J. (2014). Sessile alveolar macrophages communicate with alveolar epithelium to modulate immunity. *Nature*, *506*(7489), 503–506. <https://doi.org/10.1038/nature12902>

Wynn, T. A., & Vannella, K. M. (2016). Macrophages in Tissue Repair, Regeneration, and Fibrosis. *Immunity*, *44*(3), 450–462. <https://doi.org/10.1016/j.immuni.2016.02.015>

Dear Dr. Herold, dear Dr. Vazquez-Armendariz,

Thank you for submitting your amended manuscript (EMBOJ-2019-103476R) to The EMBO Journal. Your revised study was sent back to the three referees for re-evaluation, and we have received comments from all of them, which I enclose below. As you will see the referees find that their concerns have been sufficiently addressed and they are now broadly in favour of publication.

Thus, we are pleased to inform you that your manuscript has been accepted in principle for publication in The EMBO Journal, pending the minor remaining issues of reviewer #2 as well as additional aspects related to formatting and data representation as detailed below are addressed at re-submission.

Regarding the remaining points of referee #2, we ask you to evaluate whether you can address these with additional data or relativize claims where appropriate.

We are piloting Structured Methods a new format for the Materials and Methods of articles published at EMBO Press. Adhering to this format is optional for research articles. However, considering the strong methodological aspect of your study, we would strongly encourage you to use it. Specifically, the Material and Methods section should include a Reagents and Tools Table (listing key reagents, experimental models, software and relevant equipment and including their sources and relevant identifiers) followed by a Methods and Protocols section in which we encourage the authors to describe their methods using a step-by-step protocol format with bullet points. More information on how to adhere to this format as well as downloadable templates (.doc or .xls) for the Reagents and Tools Table can be found in the author guidelines of our sister journal Molecular Systems Biology <http://msb.embopress.org/authorguide#materialsandmethods>. An example of a paper with Structured Methods can be found here: <http://msb.embopress.org/content/14/7/e8071>. We encourage you to be even more explicit in adding details on the experimental procedures, as this should be valuable in ensuring reproducible application if the approach.

Please note that I will also follow-up shortly with a list of additional comments and edits for changes from our production team regarding statistics and data-text layout.

Please contact me at any time if you have further questions related to below points.

Thank you for giving us the chance to consider your manuscript for The EMBO Journal. I look forward to your final revision.

Again, please contact me at any time if you need any help or have further questions.

Kind regards,

Daniel Klimmeck

Daniel Klimmeck PhD
Editor
The EMBO Journal

- >> Introduce ORCID IDs for all corresponding authors (S.H.) via our online manuscript system. Please see below for additional information.
- >> Please specify distinct author contributions for Is.Sa., Ir.Sh.; Mo.He., Ma.C. He. .
- >> Rename the current 'Competing Interests' section to ' Conflict of Interest'.
- >> Please complement the Material and Methods section by indicating the animal welfare statement applicable for your study.
- >> Complement the author checklist by indicating the mice used and animal welfare statement applicable for your study.
- >> Remove the database web link from the 'Data availability" section.
- >> Please remove Dataset EV legends from the main text and zip them with each respective movie file.
- >> Adjust the reference formatting to EMBO Journal style, references in alphabetical order, ten authors et al. listed.

Please note that as of January 2016, our new EMBO Press policy asks for corresponding authors to link to their ORCID iDs. You can read about the change under "Authorship Guidelines" in the Guide to Authors here: <http://emboj.embopress.org/authorguide>

In order to link your ORCID iD to your account in our manuscript tracking system, please do the following:

1. Click the 'Modify Profile' link at the bottom of your homepage in our system.
2. On the next page you will see a box half-way down the page titled ORCID*. Below this box is red text reading 'To Register/Link to ORCID, click here'. Please follow that link: you will be taken to ORCID where you can log in to your account (or create an account if you don't have one)
3. You will then be asked to authorise Wiley to access your ORCID information. Once you have approved the linking, you will be brought back to our manuscript system.

We regret that we cannot do this linking on your behalf for security reasons. We also cannot add your ORCID iD number manually to our system because there is no way for us to authenticate this iD number with ORCID.

Thank you very much in advance.

Please see also our instructions to authors

Further information is available in our Guide For Authors:

The revision must be submitted online within 90 days; please click on the link below to submit the revision online before 18th Oct 2020.

Link Not Available

Referee #1:

The authors have adequately responded to the concerns raised by the reviewers. The additional data and revised discussion have greatly improved the manuscript.

Referee #2:

The authors addressed most of the comments. However, some clarification is needed for the following:

1. In line with my previous comment #1 and #4, the author claim that SCGB1A1+ SFTPC+ cells generate multiple lung epithelial cells in organoid models. However, except for scRNA-seq data, no

validation studies are performed to show basal cells in these organoids. Immunostaining for basal cell markers (TP63 and KRT5) is needed to substantiate their claims. Similarly, the immunostaining for Act-Tubulin is not convincing as it doesn't appear to stain cilia. This needs to be clarified.

Referee #3:

I was least critical of the three reviews based on my own expertise with these techniques. My question was adequately addressed-- defer to other two reviewers and their comments.

Point-by-point response to comments from**Referee #2:**

The authors addressed most of the comments. However, some clarification is needed for the following:

In line with my previous comment #1 and #4, the author claims that SCGB1A1+ SFTPC+ cells generate multiple lung epithelial cells in organoid models. However, except for scRNA-seq data, no validation studies are performed to show basal cells in these organoids. Immunostaining for basal cell markers (TP63 and KRT5) is needed to substantiate their claims. Similarly, the immunostaining for Act-Tubulin is not convincing as it does not appear to stain cilia. This needs to be clarified.

Response: The authors thank the reviewer for the positive feedback and the opportunity to address the remaining concerns. In addition to the scRNA-seq data, basal cells' identity within BALO's airway-like structures was confirmed by Prof. Wolfgang Kummer, an expert in lung anatomy with a focus on electron microscopy (EM) (**Figure 2I**). EM data revealed that BALO form a pseudostratified-like epithelium comprised of ciliated, secretory and basal cells, the latter presenting all the main morphological characteristics of *bona fide* basal cells including a high nucleus-to-cytoplasm ratio, being broadly attached to the basal lamina without reaching the airway lumen and with focal connections to neighboring secretory and ciliated cells (Bilodeau et al. 2014). Additionally, we have also included further confirmatory data showing a significant upregulation of the basal cell marker *p63* in day 21 BALO (**Figure 2A**) indicating that basal cell differentiation occurs over time in the developing BALO. Regarding **Figure EV2B**, we have added a new confocal picture showing a higher magnification of the β -IV tubulin⁺ ciliated cells, however, to unequivocally depict cilia we would like to refer to BALO's ultrastructure presented in **Figure 2I** which clearly shows kinocilia with the classical "9x2+2" configuration (Orhon et al. 2015).

References

- Bilodeau, Mélanie, Sharareh Shojaie, Cameron Ackerley, Martin Post, and Janet Rossant. 2014. "Identification of a Proximal Progenitor Population from Murine Fetal Lungs with Clonogenic and Multilineage Differentiation Potential." *Stem Cell Reports* 3(4):634–49.
- Orhon, I., N. Dupont, O. Pampliega, A. M. Cuervo, and P. Codogno. 2015. "Autophagy and Regulation of Cilia Function and Assembly." *Cell Death and Differentiation* 22(3):389–97.

Dear Dr. Herold, dear Dr. Vazquez-Armendariz,

Thank you for submitting the revised version of your manuscript. I have now evaluated your amended manuscript and concluded that the remaining minor concerns have been sufficiently addressed.

Thus, I am pleased to inform you that your manuscript has been accepted for publication in the EMBO Journal.

Please note that it is EMBO Journal policy for the transcript of the editorial process (containing referee reports and your response letter) to be published as an online supplement to each paper. I would thus like to ask for your consent on keeping the additional referee reply figures included in this file.

Also in case you might NOT want the transparent process file published at all, you will also need to inform us via email immediately. More information is available here:
http://emboj.embopress.org/about#Transparent_Process

Please note that in order to be able to start the production process, our publisher will need and contact you regarding the following forms:

- PAGE CHARGE AUTHORISATION (For Articles and Resources)

[http://onlinelibrary.wiley.com/journal/10.1002/\(ISSN\)1460-2075/homepage/tej_apc.pdf](http://onlinelibrary.wiley.com/journal/10.1002/(ISSN)1460-2075/homepage/tej_apc.pdf)

- LICENCE TO PUBLISH (for non-Open Access)

Your article cannot be published until the publisher has received the appropriate signed license agreement. Once your article has been received by Wiley for production you will receive an email from Wiley's Author Services system, which will ask you to log in and will present them with the appropriate license for completion.

- LICENCE TO PUBLISH for OPEN ACCESS papers

Authors of accepted peer-reviewed original research articles may choose to pay a fee in order for their published article to be made freely accessible to all online immediately upon publication. The EMBO Open fee is fixed at \$5,200 (+ VAT where applicable).

We offer two licenses for Open Access papers, CC-BY and CC-BY-NC-ND.

For more information on these licenses, please visit: <http://creativecommons.org/licenses/by/3.0/> and http://creativecommons.org/licenses/by-nc-nd/3.0/deed.en_US

- PAYMENT FOR OPEN ACCESS papers

You also need to complete our payment system for Open Access articles. Please follow this link and select EMBO Journal from the drop down list and then complete the payment process:

https://authorservices.wiley.com/bauthor/onlineopen_order.asp

On a different note, I would like to alert you that EMBO Press is currently developing a new format for a video-synopsis of work published with us, which essentially is a short, author-generated film explaining the core findings in hand drawings, and, as we believe, can be very useful to increase visibility of the work.

Please see the following link for representative recent examples:

If you have any questions, please do not hesitate to call or email the Editorial Office.

Kind regards,

Daniel Klimmeck

Daniel Klimmeck, PhD
Editor
The EMBO Journal
EMBO
Postfach 1022-40
Meyerhofstrasse 1
D-69117 Heidelberg
contact@embojournal.org
Submit at: <http://emboj.msubmit.net>

Corresponding Author Name: Susanne Herold and Ana Ivonne Vazquez Armendariz

Journal Submitted to: EMBO J

Manuscript Number: EMBOJ-2019-103476R